# Can Agents Fix Agent Issues?

**Alfin Wijaya Rahardja**∗
Fudan University
24212010055@m.fudan.edu.cn

**Junwei Liu**∗
Fudan University
jwliu24@m.fudan.edu.cn

**Weitong Chen**
Fudan University
21307130392@m.fudan.edu.cn

**Zhenpeng Chen**
Nanyang Technological University
zhenpeng.chen@ntu.edu.sg

**Yiling Lou**†
University of Illinois Urbana-Champaign
yilingl@illinois.edu

## Abstract

LLM-based agent systems are emerging as a new software paradigm and have been widely adopted across diverse domains such as medicine, robotics, and programming. However, maintaining these systems requires substantial effort, as they are inevitably prone to bugs and continually evolve to meet changing external requirements. Therefore, automatically resolving agent issues (*i.e.,* bug reports or feature requests) is a crucial and challenging task. While recent software engineering (SE) agents (*e.g.,* SWE-agent) have shown promise in addressing issues in traditional software systems, it remains unclear how effectively they can resolve real-world issues in agent systems, which differ significantly from traditional software. To fill this gap, we first manually analyze 201 real-world agent issues and identify common categories of agent issues. We then spend 500 person-hours constructing AGENTISSUE-BENCH, a reproducible benchmark comprising 50 agent issue resolution tasks (each with an executable environment and failure-triggering tests). We further evaluate state-of-the-art SE agents on AGENTISSUE-BENCH and reveal their limited effectiveness (*i.e.,* with only 0.67% - 4.67% resolution rates). These results underscore the unique challenges of maintaining agent systems compared to traditional software, highlighting the need for further research to develop advanced SE agents for resolving agent issues. Data and code are available at https://github.com/alfin06/AgentIssue-Bench.

## 1 Introduction

 LLM-based agent systems have seen widespread adoption across diverse domains, such as medicine [38, 56], programming [18, 58, 63, 61], robotics [41, 65], psychology [47, 60], and general-purpose personal assistants [16, 7]. Driven by rapid advancements, agent systems are emerging as a new software paradigm, playing an increasingly pervasive role in shaping and supporting the full spectrum of human activities.

As products of human intellectual labor, similar as traditional software systems, agent systems are also inevitably prone to quality issues. Recent work [34] has shown that multi-agent systems exhibit diverse failure modes during operation. Moreover, agent systems are continuously evolving to meet changing external requirements, making their maintenance both crucial and labor-intensive. For

---

∗Equal contribution
†Corresponding author

39th Conference on Neural Information Processing Systems (NeurIPS 2025).

instance, by May 2025, the agent system MetaGPT [26] had accumulated over 800 GitHub issues (an issue is typically a bug report or a feature request), highlighting the substantial maintenance workload associated with agent systems.

Automating the issue resolution process has been an important and challenging direction with substantial dedicated research effort. In particular, with the recent advances in agent systems, there is a growing trend toward developing software engineering agents [52, 66, 58, 18, 28, 9, 27] (referred to as *SE agents* in this paper), which can automatically resolve real-world software issues. Recent SE agents have demonstrated strong potential in resolving issues in traditional software systems. For instance, Agentless [52] correctly resolves 50.80% of issues on SWE-bench [33], a real-world issue resolution benchmark for traditional Python software.

Although SE agents have shown promise in resolving issues in traditional software systems, it remains unclear how effectively they perform on agent systems, which is a new software paradigm that differs significantly from traditional software. Therefore, in this work, we aim to answer the central question: *can **SE agents** fix issues in **agent** systems?*

To understand issues in agent systems, we first perform an empirical study to analyze and catalog real-world agent issues. In particular, we collect 201 real-world GitHub issues along with developer-committed patches from 18 widely-used agent systems. We further build a taxonomy of agent issues with human annotators via grounded theory, resulting in 6 categories and 20 sub-categories of common agent issues. Our taxonomy reveals that real-world agent systems exhibit a diverse range of issues, many of which possess unique characteristics not typically found in traditional software systems. The findings highlight the large engineering effort for maintaining agent systems, confirming that automated issue resolution for agent systems is a challenging and critical problem.

We then build AGENTISSUE-BENCH, the first *reproducible* benchmark for agent issue resolution. Reproducing agent issues is particularly more challenging compared to traditional software issues, largely due to the nondeterminism of LLMs and the volatility of external resources (*e.g.,* tools) that agents interact with. As a result, from the 201 issues analyzed, we invested 500 person-hours to successfully reproduce 50 agent issues. Each issue resolution task in AGENTISSUE-BENCH is packaged within an executable Docker environment, along with failure-triggering tests, user-reported issue descriptions, the buggy version, and the developer-committed patched version of the codebase.

We further evaluate multiple state-of-the-art SE agents (*i.e.,* Agentless [52], AutoCodeRover [66], and SWE-agent [58]) with both GPT-4o [1] and Claude-3.5-Sonnet [15] on AGENTISSUE-BENCH. We find that all of the existing SE agents exhibit limited capabilities in resolving agent issues. For instance, only 0.67% to 4.67% of agent issues are correctly resolved, which is significantly lower than the resolution rates achieved when these SE agents are applied to traditional software (*e.g.,* 23.20% - 50.80% resolution rate [33]). We further conduct a qualitative analysis to break down the resolution capabilities of SE agents across different categories. Notably, the majority of resolved issues pertain to utility or dependency issues, while the most of LLM-related issues (*e.g.,* compatibility with LLM providers or LLM operation issues) remain unsolved. Overall, our analysis reveals the limitations of current SE agents in resolving agent issues, underscoring the need for building advanced SE agents tailored to the maintenance of agent systems.

In summary, this work makes the following contributions:

- **Taxonomy.** We present the first taxonomy of issues in agent systems, derived from extensive manual analysis, which summarizes the common maintenance demands encountered during agent system evolution.

- **Reproducible benchmark AGENTISSUE-BENCH.** We manually construct the first issue resolution benchmark of real-world agent issues. Each task is packed into an executable Docker environment, including issue descriptions, failure-triggering tests, and both buggy and patched versions of the codebase, enabling easy reproduction and validation through one-click execution.

- **Evaluation.** We evaluate state-of-the-art SE agents on AGENTISSUE-BENCH with both quantitative and qualitative analysis, and find their limited capabilities in solving agent issues. Our findings highlight the unique challenges of maintaining agent systems, underscoring the need to develop more powerful SE agents for resolving agent issues.

## 2 Background and Related Work

### 2.1 LLM-based Agent Systems

LLM-based agent systems are emerging as a new software paradigm, which have been widely applied across various fields (*e.g.,* medicine [38, 56], programming [18, 58], robotics [41, 65], psychology [47, 60], and general-purpose personal assistants [16, 7]) with remarkable abilities. An LLM-based agent system [51, 50] typically consists of: (i) an LLM-controlled brain that decomposes and schedules tasks (*i.e.,* planning) and records the historical behaviors (*i.e.,* memory); (ii) a perception component that receives information from the environment; and (iii) an action component that interacts with the environment by invoking external tools. In addition, single-agent systems can collaborate to form multi-agent systems, which can tackle more complex tasks with better flexibility and effectiveness.

**Quality problems in LLM-integrated systems.** Given the widespread adoption of LLMs, recent work has been looking into quality problems (*e.g.,* bugs or runtime failures) in LLM-integrated systems. For example, Shao *et al.* [49] catalog the integration bugs in LLM and RAG systems. Different from their work, our work specifically focuses on LLM agent systems. This scope distinction leads to notable differences in taxonomies, as our taxonomy is framed from an agent architecture perspective (e.g., featuring broader coverage of tool-related issues, finer-grained categorization of memory issues, API and model binding issues). Along this direction, Cemri *et al.* [34] build a taxonomy of failure modes in multi-agent systems. While their work focuses on runtime failure symptoms by analyzing failure trajectories, our taxonomy centers on agent issue resolution by analyzing both real-world user-reported issues and developer-committed patches. Therefore, our work complements existing efforts by providing a perspective on maintaining agent systems, encompassing a broader scope that includes not only bug fixes but also feature requests. Moreover, our work is further different from existing work by introducing the first *reproducible* benchmark for agent issue resolution and empirically evaluating state-of-the-art SE agents on their ability to resolve agent issues.

### 2.2 Software Engineering Agents

Software Engineering (SE) agents are a category of agent systems specifically designed to tackle SE tasks [42]. In particular, there is a growing trend in both industry and academia toward developing SE agents [18, 28, 52, 66, 27, 58, 9, 46, 62], which can support end-to-end software maintenance by automatically resolving user-reported issues (*e.g.,* bug fixes or feature requests). For instance, Devin [18] is one of the first SE agents capable of resolving software issues by invoking file editors, terminals, and search tools. More recently, SWE-agent [58] interacts with the code repository environment through a custom Agent-Computer Interface (ACI), capable of performing actions such as manipulating files and executing bash commands; AutoCodeRover [66] incorporates a suite of code search tools that iteratively retrieve relevant code contexts to navigate the repository and localize issue locations; Moatless [27] equips agents with code search and retrieval tools to identify the issue locations; Agentless [52] optimizes the agent workflow with human expertise, incorporating hierarchical localization and regression testing to improve issue resolution rates. In this work, we evaluate the effectiveness of existing SE agents in resolving issues in agent systems.

**Benchmarking issue resolution capabilities of SE agents.** With the rise of SE agents, an increasing number of benchmarks have been developed to evaluate their capabilities in addressing real-world issue resolution tasks. For instance, Jimenez *et al.* [40] build SWE-bench from GitHub issues of 12 Python libraries. Based on SWE-bench, researchers further propose a series of benchmarks, *e.g.,* SWE-bench Lite [40], SWE-bench verified [3], and SWE-bench Lite-S [52], which are refined versions of SWE-bench with additional quality checking. While the SWE-bench series only includes issues of Python software, Zan *et al.* [64, 44] further propose SWE-bench Java, an issue resolution benchmark for Java software, and Yang *et al.* [59] build SWE-bench Multimodal, comprising frontend issue resolutions tasks from open-source JavaScript libraries. More recently, OpenAI releases SWE-Lancer Diamond [24], an issue resolution benchmark with end-to-end tests for Expensify [20] software. While existing benchmarks focus exclusively on issue resolution in traditional software systems, our work introduces the first reproducible benchmark targeting issues in agent systems, an emerging software paradigm with features distinct from traditional software. Using this benchmark, we find that current SE agents are still unable to resolve the majority of issue resolution tasks in agent systems.

# 3 Agent Issue Taxonomy

To understand issues during agent system maintenance, we first manually analyze and categorize real-world GitHub issues in widely-used agent systems.

## 3.1 Methodology

Figure 1 illustrates our methodology of systematically collecting and analyzing agent issues.

### 3.1.1 Data Collection

**Agent system collection.** To select diverse and representative agent systems, we first use the GitHub search API to obtain 50 repositories with keywords "AI agents" by Feb 2025. We then manually go through each repository to keep the ones that are LLM-based agent systems (filter out the unrelated ones like paper lists or tutorials); to focus on agent systems with active maintenance, we only keep the ones with more than 1k stars and 30 issues. In this way, we collect 18 agent systems, such as MetaGPT [26], AutoGen [11], GPT-engineer [21], and CrewAI [16]. The full list of our analyzed agent systems is in Appendix A.

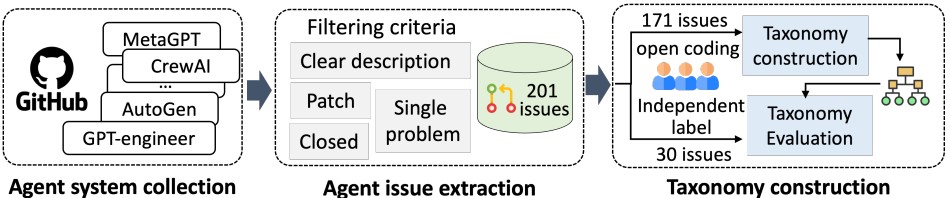

**Figure 1: Methodology of agent issue taxonomy construction.**

**Agent issue extraction.** For each studied agent system, we adopt the following inclusion criteria to extract high-quality issues. (i) The issue has been closed with a developer-committed patch to address the issue, as patches can serve as ground truth for understanding root causes of agent issues; (ii) The issue has clear descriptions without misleading information (e.g., exact patches or misleading patches in the problem description). This criteria has been widely used in constructing high-quality issue resolution benchmarks for traditional software systems [52, 3, 40]; (iii) The issue should only report one problem instead of mixing multiple problems. In the end, we obtain 201 issues in total.

### 3.1.2 Manual Labeling

We randomly separate our collected 201 agent issues into (i) 171 issues (85%) for building the taxonomy and (ii) 30 issues (15%) for evaluating our constructed taxonomy.

**Taxonomy construction.** We manually catalog the 171 agent issues with grounded theory [37]. In particular, three human annotators with extensive software development and machine learning experience apply open coding [35, 36, 48] to annotate each issue based on the issue description and the developer-committed patch. They break down each issue into segments and label them with descriptive codes. Then they organize the open codes into structured categories by merging and linking relevant ones. All the annotators further discuss and review the taxonomy until reaching a consensus.

**Taxonomy evaluation.** We further evaluate our taxonomy on the remaining 30 agent issues. Two annotators independently label each issue. Their annotation reaches a high agreement ratio (Cohen's Kappa = 0.849); meanwhile there emerge no new categories in addition to our taxonomy during their annotation.

## 3.2 Taxonomy

Table 1 presents our taxonomy of agent issues, mainly covering 6 categories. Appendix F presents detailed examples for each sub-category. In addition to the *"utility issues"* category which may also occur in traditional software systems, the remaining five categories are uniquely tied to key agent system components (*e.g.,* tools and memory), making them distinctive to agent systems.

Table 1: Taxonomy of agent issues.

| Category | Sub-category | Description |
|---|---|---|
| Incompatibility with LLM providers (7.46%) | Incompatible dependencies (1.49%) | Miss or misuse the libraries of LLM providers. |
| | Unsupported models (2.99%) | Lack the support of recent LLMs. |
| | Incompatible parameters to LLM providers (2.99%) | Use undefined parameters or miss parameters of LLM query interfaces. |
| Tool-related issues (19.90%) | Tool dependency issues (3.48%) | Miss or misuse libraries for running agent-invoked tools. |
| | Tool configuration issues (3.47%) | Misconfigure the settings of agent-invoked tools. |
| | Tool implementation errors (8.46%) | Incorrect implementation of self-developed agent-invoked tools. |
| | Misuse tool interfaces (4.48%) | Incorrect tool invocation due to missing/wrong parameters. |
| Memory-related issues (14.43%) | Memory initialization issues (2.49%) | Incomplete or inconsistent memory states due to database initialization or workspace resetting issues. |
| | Memory content errors (10.95%) | Incorrect message attributes, misleading content, or content loss caused by faulty storage logic. |
| | Memory dependency issues (1.00%) | Incorrect internal module dependencies or external libraries required by memory operations. |
| LLM operation issues (31.34%) | Model access misconfiguration (6.97%) | Model access errors caused by misconfiguration like incorrect model binding or authentication credentials (*e.g.,* API keys). |
| | Token usage misconfiguration (3.48%) | LLM token management issues such as incorrect limits or pricing. |
| | Incorrect model output handlers (8.46%) | Incorrect parsing logic for model output or miss handlers for unexpected model behaviors like empty or exceptional responses. |
| | Model dependency issues (2.99%) | Missing/incompatible libraries related to model operation such as tokenization or transformer dependency conflicts. |
| | Context length issues (4.98%) | Truncated outputs caused by exceeding context limits or miscalculating context length. |
| | Prompt-related issues (4.48%) | Suboptimal prompt content or prompt management issues (*e.g.,* fail to set/update prompts). |
| Workflow issues (6.47%) | | Abnormal agent workflows like hanging or repeated loops. |
| Utility issues (20.40%) | Utility implementation issues (8.96%) | Implementation errors in LLM-unrelated components (e.g., UI/Docker/logging). |
| | Utility dependency issues (4.48%) | Miss/incompatible libraries or circular internal dependencies required by general utilities (*e.g.,* testing or file operations). |
| | Utility configuration issues (6.97%) | External component misconfiguration (*e.g.,* I/O paths, network settings). |

*Incompatibility with LLM providers.* Most agent systems incorporate existing LLMs from LLM providers (*e.g.,* OpenAI [2], DeepSeek [17], and Anthropic [10]), and improper usage of providers' interfaces impairs agent functionality. Such issues often stem from missing dependencies or incorrect invocations of provider APIs. Moreover, due to the rapid evolution of LLMs, users frequently request new feature to support newly-released LLMs.

*Tool-related issues.* The versatility of agent systems partly stems from their proficiency in utilizing tools to interact with the environment. As a result, many agent-related issues arise during tool invocation, including missing tool-dependent libraries, misconfigurations, or incorrect use of tool interfaces. In addition to external tools, agents may also rely on internal tools (*e.g.,* custom-developed functions), where implementation flaws can trigger unintended behaviors during tool execution.

*Memory-related issues.* The memory mechanism in agents tracks the trajectory of agent operation, and most memory-related issues arise from incorrect memory content. For example, agents may pollute memory with irrelevant information when they mistakenly extract unrelated attributes from the current context, or memory entries may be missing or incomplete due to failures in storing data.

*Workflow issues.* Due to the autonomy and flexibility of agent systems, unexpected behaviors can emerge along the agent workflow, such as repeated actions or hanging states. Although it is difficult to completely eliminate such issues, developers commonly mitigate them by incorporating status checkers to monitor and regulate the agent workflow.

*LLM operation issues.* A large portion (31.34%) of agent-related issues occur during LLM operation. For example, proper configuration of model access and token usage is critical, and misconfiguration in these areas can disrupt agent functionality. Additionally, many issues stem from incorrect handling of model outputs, including: (i) flawed parsing implementations, or (ii) missing handlers for unexpected model responses. Beyond the suboptimal prompt content (*e.g.,* unclear model instructions), prompt management can also introduce risks: as agent systems often maintain a large and evolving pool of prompts, failures in prompt updates or configuration can result in models being queried with incorrect or outdated instructions.

**Summary.** Our taxonomy reveals that real-world agent systems exhibit a diverse range of issues, many of which possess unique characteristics not typically found in traditional software systems. In particular, developing and maintaining agent systems demands substantial engineering effort, as developers must manage correct dependencies, configurations, and implementations across multiple components (*e.g.,* model providers, LLM operations, memory mechanisms, and tools). Therefore, we believe that automatically resolving issues in agent systems represents a challenging and increasingly vital research direction in the era of LLMs.

# 4 AGENTISSUE-BENCH Benchmark

We then manually build AGENTISSUE-BENCH, the first *reproducible* issue resolution benchmark of real-world agent issues. AGENTISSUE-BENCH can be used to evaluate the efficacy of state-of-the-art SE agents in solving issues in agent systems.

## 4.1 Benchmark Construction

We construct AGENTISSUE-BENCH out of the 201 GitHub agent issues we collected in Section 3. In particular, we try to reproduce each issue according to the following procedure.

**Step 1: Failure reproduction.** For each issue, we pull its corresponding buggy commit and set up the agent system. In particular, we manually write a test script (*i.e.,* failure-triggering test) to reproduce the problematic behaviors according to the issue descriptions. In this step, we filter out the issues where we cannot observe the same buggy behavior as issue descriptions.

**Step 2: Patch reproduction.** We then pull the corresponding patched commit and execute the failure-triggering test on it. In this step, we only keep the issues where the patched version can pass the failure-triggering tests (*i.e.,* problematic behaviors disappear on the patched version).

**Step 3: Non-flakiness verification.** Given the nondeterminism of LLMs, we repeat the previous two steps three times for each issue so as to eliminate the test flakiness. In this step, we filter out issues where there are inconsistent behaviors on executing one failure-triggering test.

Through such a multi-step filtering process, the original 201 agent issues are narrowed down to 50 reproducible issue resolution tasks, collectively forming AGENTISSUE-BENCH. We find that reproducing issues in agent systems is significantly more challenging than in traditional software systems, as agent issues are associated with diverse internal and external components and resources. In particular, most agent issues fail to reproduce for the following reasons. (i) The nondeterminism of LLMs leads to unstable model outputs, which hinders the reproduction of agent issues such as workflow errors; (ii) External resources (*e.g.,* agent-invoked tools, dependent libraries, or LLM providers) may have changed since the issue was reported, making it impossible to reproduce the same failure; (iii) Issue descriptions lack sufficient details or steps on how to reproduce the problematic behaviors; (iv) Agent systems cannot be correctly set up and exhibit unexpected failure behaviors that are different from the issue descriptions. Overall, the entire reproduction process takes huge manual effort (approximately 500 person-hours).

## 4.2 Benchmark Details

**Benchmark statistics.** Figure 2 shows the distribution of AGENTISSUE-BENCH across different issue categories. Overall, we can observe that the 50 reproduced agent issues in AGENTISSUE-BENCH cover all the main categories identified in our taxonomy of agent issues, indicating that AGENTISSUE-BENCH is representative of real-world agent issue distribution. Moreover, issues in AGENTISSUE-BENCH involve patches of different scales (Detailed statistics are in Table 5).

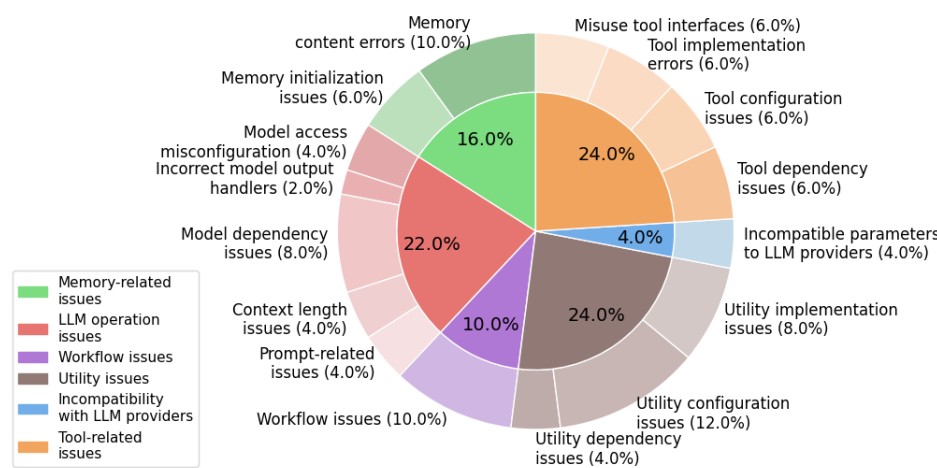

Figure 2: Distribution of AGENTISSUE-BENCH

Each issue resolution instance in AGENTISSUE-BENCH consists of the following components: (i) *Issue description:* a user-reported textual description of the problem; (ii) *Buggy version of the agent system:* the buggy commit of the agent code repository in which the issue occurs; (iii) *Developer-committed patch:* the code changes between the buggy and correct versions, serving as the ground truth for issue resolution; (iv) *Failure-triggering tests:* test scripts that reproduce the issue on the buggy version but pass on the patched version; (v) *Docker environment*: a container with all necessary dependencies and configurations to execute the agent system.

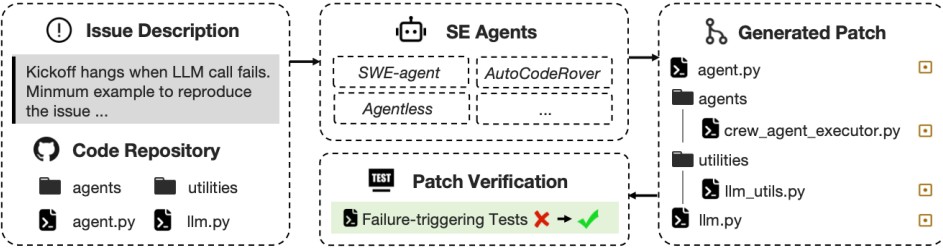

Figure 3: A task example in AGENTISSUE-BENCH.

**Task formulation.** The agent issue resolution task can be formulated as follows: (i) *Input:* the issue description and the buggy codebase of the agent system; (ii) *Output:* a patch (*i.e.,* a code edit to the buggy codebase) that aims to resolve the issue. Figure 3 shows the task example in AGENTISSUE-BENCH.

**Evaluation metrics.** To evaluate how a technique tackles the agent issue resolution task, we adopt the following metrics to evaluate the patches output by the technique (*i.e.,* SE agents in our experiments). (i) *Localization accuracy*: if the generated patch modifies the same location as the developer-committed patch, we consider it to have accurately localized the issue. We then compute the percentage of issues for which the generated patches can achieve accurate localization. (ii) *Plausible resolution rate*: if the generated patch makes the failure-triggering tests pass after being applied, we consider it to plausibly resolve the issue (*i.e.,* denoted as a plausible patch). We then compute the percentage of issues for which the generated patches are plausible patches. (iii) *Correct resolution rate*: if the generated plausible patch is further semantically-equivalent to the developer-committed patch, we consider it to correctly resolve the issue (*i.e.,* denoted as a correct patch). In particular, given the insufficiency of tests in practice, it is common [45, 57, 43] that plausible patches are not necessarily correct patches but are just overfitting to the failure-triggering tests. Therefore, only reporting the plausible resolution rate can overestimate the effectiveness of issue resolution techniques. Following the common practice in the program repair area [55, 54, 53, 39], we

Table 2: Overall results of SE agents on AGENTISSUE-BENCH

| SE Agent | LLM | Plausibly resolved% | Correctly resolved% | Localization % File-level | Localization % Function-level | Avg. $Cost |
|---|---|---|---|---|---|---|
| Agentless | GPT-4o | 12.00 | 3.33 | 27.82 | 12.99 | 0.65 |
|  | Claude-3.5-S | 12.00 | 4.00 | 27.35 | 17.50 | 0.33 |
| AutoCodeRover | GPT-4o | 7.33 | 1.33 | 22.07 | 14.77 | 0.23 |
|  | Claude-3.5-S | 12.67 | 4.67 | 25.81 | 19.18 | 0.05 |
| SWE-agent | GPT-4o | 0.67 | 0.67 | 11.67 | 4.22 | 1.15 |
|  | Claude-3.5-S | 2.00 | 2.00 | 9.52 | 6.78 | 0.57 |

further involve human annotators to manually check whether the plausible patches are semantically equivalent to developer-committed patches. In particular, two participants independently review each plausible patch by comparing it to the golden patch (i.e., developer-committed), focusing on whether the semantics of the patch fully resolve the underlying issue as intended and do not introduce other functional or semantic errors. If both reviewers agree that the patch is semantically equivalent and correctly resolved the issue, it is labeled as correct. If there is a disagreement between the two reviewers, a third participant would be involved as an adjudicator. The final label is determined only after all three reviewers reach a consensus. We then compute the percentage of issues for which the generated patches are correct patches.

## 5 Experiments

In this section, we investigate how state-of-the-art SE agents can automatically resolve real-world issues in agent systems by evaluating their efficacy on AGENTISSUE-BENCH.

### 5.1 Experimental Setup

**Studied SE agents.** We include three state-of-the-art SE agents, including SWE-agent [58], AutoCodeRover [66], and Agentless [52]. These agents are selected given that they are fully open-sourced and achieve superior effectiveness in resolving issues for traditional software systems [33]. We directly adopt their released implementation with the original hyperparameter settings.

**Backbone LLMs.** Based on the recent SWE leaderboard [33], state-of-the-art SE agents achieve higher fixing rate on general software issues when equipped with backbone LLMs GPT-4o [1] and Claude-3.5 Sonnet [15]. Therefore, in our experiments, we mainly study how effective SE agents are in resolving agent issues with these two backbone LLMs (temperature = 0).

**Evaluation pipelines.** We apply studied SE agents on AGENTISSUE-BENCH and collect their generated patches for each issue resolution task. We then calculate the metrics of fault localization accuracy, plausible and correct resolution rates for each studied SE agent. To eliminate the randomness from LLMs, we repeat all experiments three times and present the average results. In particular, our major metric (average resolution rate over three runs) is essentially average pass@1 over three runs. Table 8 in Appendix G further presents the pass@1 and pass@3 over one run.

### 5.2 Quantitative Results

**Overall resolution effectiveness.** Table 2 shows the results of the studied SE agents on AGENTISSUE-BENCH. In general, state-of-the-art SE agents can only correctly resolve a small number (*i.e.,* 0.67% - 4.67%) of agent issues. In addition, in most cases, SE agents even fail to correctly identify the location (*i.e.,* files or functions) for resolving the issue, *e.g.,* file-level/function-level localization accuracy is less than 28%/20%. Such observations reveal the limited capabilities of state-of-the-art SE agents in understanding and resolving the issues in agent systems.

In addition, Figure 4 compares the correct resolution rate of SE agents on agent issues (on our benchmark AGENTISSUE-BENCH) versus on traditional software issues (results from SWE-bench Lite [33]). As there is no previous data of AutoCodeRover with Claude-3.5-S on SWE-bench, we leave it as blank. Overall, SE agents demonstrate significantly lower resolution rates on agent issues compared to traditional software issues. These findings highlight the unique challenges posed by agent systems and underscore the need for developing SE agents specifically tailored to maintain agent systems, which is an emerging and distinctive software paradigm.

Table 3: Breakdown of resolved agent issues (unresolved categories are not presented).

| Category | Resolved% | Sub-category | Resolved% |
|---|---|---|---|
| Tool-related issues | 2/12 (16.67%) | Tool dependency issues | 2/3 (66.67%) |
| LLM operation issues | 1/11 (9.09%) | Prompt-related issues | 1/2 (50.00%) |
| Utility issues | 2/12 (16.67%) | Utility configuration issues | 2/6 (33.33%) |

**Comparison among SE agents and backbone LLMs.** As shown in Table 2, SE agents with Claude-3.5-S achieve better effectiveness than with GPT-4o in terms of plausible resolution, correct resolution, and localization accuracy. In particular, AutoCodeRover with Claude-3.5-S achieves the highest resolution rate (*i.e.,* 4.67%) and the highest function-level localization accuracy (*i.e.,* 19.18%). Overall, we observe a larger potential of Claude-3.5-S in understanding agent issues than GPT-4o.

Figure 5 shows the unique and overlapped agent issues that are correctly resolved by each SE agent. An issue is counted as correctly resolved by an agent if it was solved in at least one of the three experimental runs. We could observe that each SE agent can uniquely fix 1 - 2 bugs that cannot be resolved by any other SE agents. In addition, there is no agent issue that can be fixed by all SE agents. In other words, existing SE agents exhibit complementary capabilities to resolve agent issues.

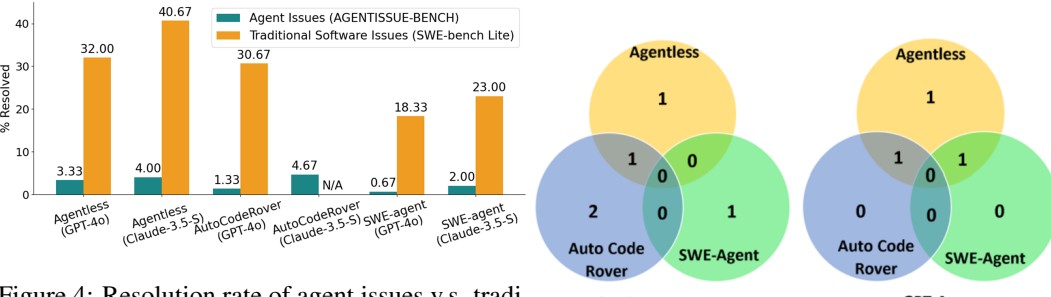

Figure 4: Resolution rate of agent issues v.s. traditional software issues.

Figure 5: Venn diagrams of resolved issues.

**Costs.** As shown in Table 2, the average costs of applying SE agents to agent issue are controllable, ranging from $0.05 to $1.15. The cost range is similar as applying these SE agents to resolve traditional software issues (*e.g.,* $0.45 - $2.53 [52]).

## 5.3 Qualitative Results

In this section, we further break down the issues that SE agents can and cannot resolve, aiming to better understand their strengths and limitations in resolving agent issues. Table 3 presents the issue categories that can be resolved by at least one studied SE agent.

**Resolved agent issues.** Overall, the majority of agent issues resolved by SE agents are still related to utility (*e.g.,* log/file operation/UI), which actually share high commonality with traditional software systems. As a result, SE agents are inherently able to resolve issues of this category in agent systems. Moreover, besides common utility issues, some of the dependency issues on agent-specific components (*e.g.,* tool) can also be resolved by SE agents. The reason why SE agents can handle such agent issues might be that the dependency issues often contain explicit error messages (*e.g.,* missing libraries or incompatible variables/interfaces). As a result, even if the dependencies are unique to agent components (*e.g.,* tool), they can still be similar to dependency issues in other general software components, which are straightforward and informative to resolve.

**Unresolved agent issues.** Overall, the majority of agent-specific issues cannot be resolved by any SE agent. For example, SE agents resolve a very few (or even none) issues on LLM provider incompatibility, memory, or LLM operation. The reason might be that the exchanges with LLM providers are unique features in agent systems and agent systems are emerging in the recent period, which thus are less covered in the LLM training data. In addition, the autonomous and flexible nature of agent systems stemming from LLMs makes it challenging to identify the root causes of LLM operation issues. Figure 6 and Figure 7 in Appendix E show two unresolved issues for which all SE agents cannot even correctly localize the buggy files.

In summary, our analysis further confirms the limitations of existing SE agents in resolving the agent issues which are particularly related to agent-specific features, highlighting the necessity of building more advanced SE agents for maintaining agent systems.

## 6 Limitation and Future work

While AGENTISSUE-BENCH is representative of real-world agent issues by covering a wide range of different categories, the generality of our findings can still be restricted due to the data source and the benchmark scale. First, our work only focuses on reactive agent issues (i.e., first-expose-then-fix), as our data source (i.e., user-reported Github issues) inherently captures problems reported by agent users. This scope intentionally excludes other maintenance aspects such as preventative strategies (e.g., proactive LLM monitoring) and performance optimization, which are typically observed from the developers' perspective using internal logs. Second, the current benchmark size is limited as reproducing issues in agent systems is significantly more challenging than in traditional software systems. Due to the nondeterminism of LLMs and changeable external resources (*e.g.,* tools and LLM providers) interacted with agent systems, only a small number of agent issues (50 out of 201 issues) can be successfully reproduced. Moreover, huge manual effort (approximately 500 person-hours) are dedicated to preparing the Docker environment, configuring agent systems, and writing failure-triggering tests. In the future, we plan to continuously maintain and extend our benchmark to support future research on agent system maintenance. The continuous work of our benchmark is available at our website [6]. In particular, the benchmark has been extended with 20 more reproducible issues since the paper submission time. Similar trends (i.e., poor resolution rate) can be observed in those additional issues. Detailed results are presented in Appendix H.

**Discussion.** Based on our findings, we further discuss implications for future research towards building more effective SE agents for resolving agent issues. (i) *Adding a knowledge base on agent-needed external resources.* Our findings show that existing SE agents struggle with issues related to external resources. A promising direction is to augment agents with an evolving knowledge base built from API documentation, release notes, and historical issues. Integrating this knowledge could empower SE agents to better reason about and diagnose the issues related to external resources. (ii) *Training SE agents with instances and trajectories collected from* AGENTISSUE-BENCH. Our benchmark and study provide training data specifically on the emerging agent systems. As our work provides executable environments and tests of buggy/fixed agent systems, future work can collect instances and trajectories (e.g., agent-environment/tool interactive trajectories) for fine-tuning more powerful SE agents that specifically targets at agent issue resolution. (iii) *Adding a dynamic analysis component in SE agents.* Our results highlight the limited localization accuracy of current agents, suggesting a large gap between an issue description and its root cause. To address this, future SE agent architectures could move beyond static analysis and incorporate a dynamic analysis component. By utilizing runtime information like execution trajectories and tool outputs, the agent can gather richer signals for more accurate bug localization and patch generation.

## 7 Conclusion

In this work, we analyze 201 GitHub issues from 18 real-world agent systems and construct the first taxonomy of agent issues. We further build AGENTISSUE-BENCH, the first *reproducible* benchmark of 50 high-quality agent issue resolution tasks. Experiments on state-of-the-art SE agents demonstrate their limited effectiveness in addressing agent issues (with resolution rates ranging from 0.67% to 4.67%), highlighting the unique challenges in maintaining agent systems and the pressing need for more advanced SE agents tailored to this emerging software paradigm.

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

# A   Analyzed agent systems

Table 4 presents the full list of our analyzed agent systems and their statistics.

Table 4: Statistics of analyzed agent systems

| Agent | #stars | #Loc | Creation time | License |
|---|---|---|---|---|
| agent-squad [4] | 5.4k | 109,019 | 07/23/2024 | Apache-2.0 License |
| AGiXT [7] | 3k | 111,946 | 04/04/2023 | MIT License |
| AI SDK [8] | 18.4k | 365,300 | 05/23/2023 | Apache-2.0 License |
| autogen [11] | 44.2k | 197,969 | 08/18/2023 | CC-BY-4.0, MIT License |
| camel [13] | 12.4k | 206,152 | 03/17/2023 | Apache-2.0 License |
| babyagi [12] | 21.4k | 8,800 | 04/03/2023 | MIT License |
| CrewAI [16] | 31.3k | 171,395 | 10/27/2023 | MIT License |
| Haystack [23] | 22.9k | 180,500 | 11/14/2019 | Apache-2.0 License |
| Lagent [25] | 2.1k | 13,075 | 08/20/2023 | Apache-2.0 License |
| MetaGPT [26] | 55.4k | 90,709 | 06/30/2023 | MIT License |
| RagaAI-Catalyst [30] | 16.2k | 47,252 | 10/01/2024 | Apache-2.0 License |
| ChatDev [14] | 26.8k | 40,478 | 11/04/2023 | Apache-2.0 License |
| gpt-engineer [21] | 54.1k | 17,460 | 04/29/2023 | MIT License |
| Pythagora [29] | 1.8k | 5,859 | 01/21/2023 | Apache-2.0 License |
| SWE-agent [32] | 15.7k | 63,388 | 04/02/2024 | MIT License |
| evo.ninja [19] | 1.1k | 31,862 | 08/18/2023 | MIT License |
| Superagent [31] | 5.8k | 58,602 | 05/10/2023 | MIT License |
| gpt-researcher [22] | 21.3k | 168,849 | 05/12/2023 | Apache-2.0 License |

# B   Patch Scales of AGENTISSUE-BENCH

Table 5 presents the statistics of the patch scales in AGENTISSUE-BENCH.

# C   Human Participation

All human-involved tasks in our experiments (including taxonomy construction, taxonomy evaluation, and issue reproduction) were approved by the Institutional Review Board (IRB) at our institution. Additionally, all participants were compensated at a rate of $15 per hour.

In taxonomy evaluation, each human annotator is provided with the following instructions: *"Given the taxonomy of agent issues (along with the definition of each agent issue category), please label each agent issue with any category in the taxonomy. If there are no applicable categories in the given taxonomy, please return the label as non-applicable."*

# D   Experiment Statistical Significance

Table 6 presents the mean results of SE agents on AGENTISSUE-BENCH, along with their corresponding two-sigma ($\pm 2\sigma$) errors. To calculate the two-sigma errors, we conducted the experiment on AGENTISSUE-BENCH three times, computed the standard deviation for the results and multiplied it by 2, as shown in Equation 1:

$$2\sigma = 2 \times \sqrt{\frac{1}{N-1} \sum_{i=1}^{N}(x_i - \bar{x})^2} \tag{1}$$

where $N$ is the number of experimental runs, $x_i$ is the result of the $i$-th run, and $\bar{x}$ is the mean result.

# E   Examples of Unresolved Issues

In this section, we provide two issue examples that all SE agents fail to localize the buggy files and generate correct patches.

For the example depicted in Figure 6, the agent lacks up-to-date knowledge regarding which LLMs currently support the "stop" parameter. As a result, the agent incorrectly passed the "stop" parameter to LLMs that do not support it (*i.e., o1-preview* and *o1-mini*), ultimately aggravating the issue.

Table 5: Mean and maximum values for various patch attributes in studied agents

| Attribute | Mean | Max |
|---|---|---|
| # Lines edited | 66.05 | 355 |
| # Files edited | 3.58 | 34 |
| # Functions edited | 6.79 | 54 |

Table 6: Mean results with 2-sigma($\pm2\sigma$) errors of SE agents on AGENTISSUE-BENCH

| SE Agent | LLM | Plausibly resolved % | Correctly resolved % | Localization % File-level | Function-level |
|---|---|---|---|---|---|
| Agentless | GPT-4o | 12.00 ($\pm$4.00) | 3.33 ($\pm$4.62) | 27.82 ($\pm$5.51) | 12.99 ($\pm$2.78) |
| | Claude-3.5-S | 12.00 ($\pm$0.00) | 4.00 ($\pm$0.00) | 27.35 ($\pm$0.00) | 17.50 ($\pm$0.00) |
| AutoCodeRover | GPT-4o | 7.33 ($\pm$2.31) | 1.33 ($\pm$2.31) | 22.07 ($\pm$7.21) | 14.77 ($\pm$2.62) |
| | Claude-3.5-S | 12.67 ($\pm$6.11) | 4.67 ($\pm$2.31) | 25.81 ($\pm$11.43) | 19.18 ($\pm$5.54) |
| SWE-agent | GPT-4o | 0.67 ($\pm$2.31) | 0.67 ($\pm$2.31) | 11.67 ($\pm$5.17) | 4.22 ($\pm$4.07) |
| | Claude-3.5-S | 2.00 ($\pm$0.00) | 2.00 ($\pm$0.00) | 9.52 ($\pm$5.24) | 6.78 ($\pm$2.59) |

For the example depicted in Figure 7, the agent fails to identify the root cause of the *KeyError*, *i.e.,* a conflict arising from generating multiple diffs for a single file. This issue is specific to the agent system, as it involves the handling of model outputs. However, instead of performing a deeper analysis, the agent merely prints an error message, resulting in an unsuccessful patch.

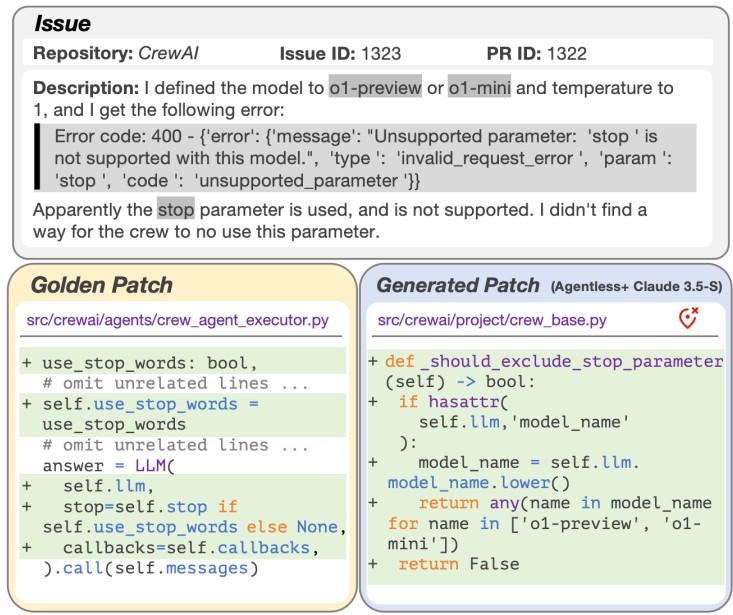

Figure 6: This unresolved issue arises because not all LLMs support the 'stop' parameter, requiring users to control its use (*e.g.,* via *use_stop_words* in the Golden Patch). The agent-generated patch aggravated the issue by passing 'stop' to unsupported models (*i.e., o1-preview* and *o1-mini*).

## F   Examples of Issues in Different Categories

In this section, we provide detailed issue examples of each sub-category in Table 1. For each issue, we provide the repository name, the user-provided issue description and the summarization of the developer-committed patch (along with the link to the original issue and Pull Request (PR) pages).

### F.1   Incompatibility with LLM providers

1. **Incompatible dependencies**
   - **Repository**: *gpt-researcher*
   - **Link to the Issue**: https://github.com/assafelovic/gpt-researcher/issues/1106

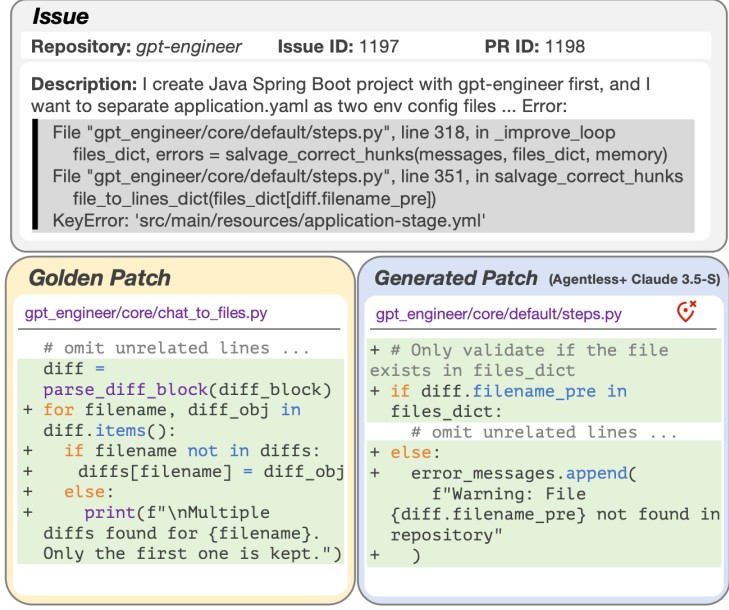

**Issue**

**Repository:** *gpt-engineer*        **Issue ID:** 1197        **PR ID:** 1198

**Description:** I create Java Spring Boot project with gpt-engineer first, and I want to separate application.yaml as two env config files ... Error:

```
File "gpt_engineer/core/default/steps.py", line 318, in _improve_loop
    files_dict, errors = salvage_correct_hunks(messages, files_dict, memory)
File "gpt_engineer/core/default/steps.py", line 351, in salvage_correct_hunks
    file_to_lines_dict(files_dict[diff.filename_pre])
KeyError: 'src/main/resources/application-stage.yml'
```

**Golden Patch**

`gpt_engineer/core/chat_to_files.py`

```
  # omit unrelated lines ...
  diff =
  parse_diff_block(diff_block)
+ for filename, diff_obj in
  diff.items():
+   if filename not in diffs:
+     diffs[filename] = diff_obj
+   else:
+     print(f"\nMultiple
  diffs found for {filename}.
  Only the first one is kept.")
```

**Generated Patch**  (Agentless+ Claude 3.5-S)

`gpt_engineer/core/default/steps.py`

```
+ # Only validate if the file
exists in files_dict
+ if diff.filename_pre in
  files_dict:
    # omit unrelated lines ...
+ else:
+   error_messages.append(
      f"Warning: File
  {diff.filename_pre} not found in
  repository"
+   )
```

Figure 7: This unresolved issue stems from the LLM generating multiple diffs for a single file, resulting in conflicts. The Golden Patch resolves this by retaining only the first diff. In contrast, the agent-generated patch fails to investigate the root cause and simply logs an error message.

- **Link to the PR**: https://github.com/assafelovic/gpt-researcher/pull/1161
- **Issue Description**: Testing revealed that the invocation method for `token_counter` and related functions in Claude has changed, requiring verification.
- **Fix Strategy**: Update the version of *anthropic* library and use the latest released APIs.

2. **Unsupported models**
   - **Repository**: *ChatDev*
   - **Link to the Issue**: https://github.com/OpenBMB/ChatDev/issues/284
   - **Link to the PR**: https://github.com/OpenBMB/ChatDev/pull/277
   - **Issue Description**: Can't do anything with 3.5 turbo. The code it makes is brutal. Can it be possible to add GPT 4 Turbo? gpt-4-1106-preview
   - **Fix Strategy**: Update the version of *openai* library and add support for GPT-4 Turbo.

3. **Incompatible parameters to LLM providers**
   - **Repository**: *CrewAI*
   - **Link to the Issue**: https://github.com/crewAIInc/crewAI/issues/1323
   - **Link to the PR**: https://github.com/crewAIInc/crewAI/pull/1322
   - **Issue Description**: I defined the model to o1-preview or o1-mini and temperature to 1, and I get the following error. `Unsupported parameter: "stop" is not supported with this model.` Apparently the stop parameter is used, and is not supported. I didn't find a way for the crew to no use this parameter.
   - **Fix Strategy**: Added the option `use_stop_words` to allow users to configure whether to use the `stop` parameter.

## F.2    Tool-related issues

1. **Tool dependency issues**
   - **Repository**: *lagent*
   - **Link to the Issue**: https://github.com/InternLM/lagent/issues/279
   - **Link to the PR**: https://github.com/InternLM/lagent/pull/280

- **Issue Description**: When running the agent with web search capabilities, an error occurred. `ModuleNotFoundError: No module named "tenacity"`

- **Fix Strategy**: Add `tenacity` in the dependency configuration file.

2. **Tool configuration issues**
   - **Repository**: *gpt-researcher*
   - **Link to the Issue**: https://github.com/assafelovic/gpt-researcher/issues/922
   - **Link to the PR**: https://github.com/assafelovic/gpt-researcher/pull/925
   - **Issue Description**: It looks like we cannot set RETRIVER solely to duckduckgo or others. It always throws an exception about `Exception: Tavily API key not found. Please set the TAVILY_API_KEY environment variable.`
   - **Fix Strategy**: Ensure the retriever is set up according to the user's configuration specified via environment variables.

3. **Tool implementation issues**
   - **Repository**: *SWE-agent*
   - **Link to the Issue**: https://github.com/SWE-agent/SWE-agent/issues/697
   - **Link to the PR**: https://github.com/princeton-nlp/SWE-agent/pull/731
   - **Issue Description**: If the agent tries to `cat` out the content of a word file (.docx), then line `buffer.decode()` fails, and the program crashes.
   - **Fix Strategy**: Replace `buffer.decode()` with `buffer.decode("utf-8",errors="backslashreplace")` so that the program will not crash when reading non-utf8 encoded bytes.

4. **Misuse tool interfaces**
   - **Repository**: *camel*
   - **Link to the Issue**: https://github.com/camel-ai/camel/issues/256
   - **Link to the PR**: https://github.com/camel-ai/camel/pull/258
   - **Issue Description**: When "KAUST" is the entity word to be searched, the returned result by wikipedia API (wikipedia.summary) is the summary about KAIST.
   - **Fix Strategy**: Set the `auto_sugget` parameter to `False` when invoking `wikipedia.summary()` so that it does not change the search word (such as KAUST -> KAIST)

## F.3 Memory-related issues

1. **Memory initialization issues**
   - **Repository**: *CrewAI*
   - **Link to the Issue**: https://github.com/crewAIInc/crewAI/issues/2123
   - **Link to the PR**: https://github.com/crewAIInc/crewAI/pull/2182
   - **Issue Description**: Looks like `reset-memories` is throwing an error on `-a`. `An unexpected error occurred: No crew found.`
   - **Fix Strategy**: Fix the `get_crew` method to obtain the correct `crew` instance, and ensure that `memory` is only reset when it is not `None`.

2. **Memory content issues**
   - **Repository**: *camel*
   - **Link to the Issue**: https://github.com/camel-ai/camel/issues/915
   - **Link to the PR**: https://github.com/camel-ai/camel/pull/916
   - **Issue Description**: Current (memory storage) logic checks the content in the chunk, if the content is `None` then the message would be appended, but for some API like SambaNova, there may include many None content in chunks in the middle of response. We need to change the logic, checking `choice.finish_reason` then append the message would be better.
   - **Fix Strategy**: Update the logic for storing messages. Determine whether a chunked message has been fully stored by checking if the `finish_reason` attribute is not `None`.

3. **Memory dependency issues**
   - **Repository**: *autogen*
   - **Link to the Issue**: https://github.com/microsoft/autogen/issues/4245
   - **Link to the PR**: https://github.com/microsoft/autogen/pull/4246
   - **Issue Description**: Running `autogenstudio ui –port 8081` fails with `ImportError: cannot import name 'InnerMessage' from 'autogen_agentchat.messages'`
   - **Fix Strategy**: Since 'InnerMessage' has been renamed to 'AgentMessage', all references to 'InnerMessage' are renamed to 'AgentMessage'.

## F.4   LLM operation issues

1. **Model access misconfiguration**
   - **Repository**: *camel*
   - **Link to the Issue**: https://github.com/camel-ai/camel/issues/1273
   - **Link to the PR**: https://github.com/camel-ai/camel/pull/1277/commits
   - **Issue Description**: The decorator `api_keys_required()` currently only supports setting the value of "DUMMY_TOKEN" in the environment variable, but does not support directly calling `DummyClass(api_key="xxxx")`.
   - **Fix Strategy**: Refactor the `api_keys_required()` decorator and make it compatible with the method of directly setting API key.

2. **Token usage misconfiguration**
   - **Repository**: *camel*
   - **Link to the Issue**: https://github.com/camel-ai/camel/issues/1018
   - **Link to the PR**: https://github.com/camel-ai/camel/pull/1071
   - **Issue Description**: For LLM API served by OpenAI-compatible providers, if the `max_tokens` is not provided then it would use `NOT_GIVEN` from openai, which will lead to `TypeError: '<=' not supported between instances of 'int' and 'NotGiven'`
   - **Fix Strategy**: Unify the default `token_limit` property in the base model class to make sure it is provided for different models.

3. **Incorrect model output handlers**
   - **Repository**: *MetaGPT*
   - **Link to the Issue**: https://github.com/geekan/MetaGPT/issues/1100
   - **Link to the PR**: https://github.com/geekan/MetaGPT/pull/1105
   - **Issue Description**: When running `python3 debate.py "Talk about Artificial General Intelligence"`, an error occurs: `ValueError: The response.text quick accessor only works for simple (single-Part) text responses`.
   - **Fix Strategy**: The root cause is that the Gemini model flags the request as potentially involving sensitive or harmful content and blocks it. To address this scenario, the `BlockedPromptException` has been added to catch exceptions triggered by blocked prompts.

4. **Model dependency issues**
   - **Repository**: *lagent*
   - **Link to the Issue**: https://github.com/InternLM/lagent/issues/244
   - **Link to the PR**: https://github.com/InternLM/lagent/pull/245
   - **Issue Description**: Encounter `AttributeError: 'GenerationConfig' object has no attribute '_eos_token_tensor'` when running code in the *transformers* library.
   - **Fix Strategy**: Update the version constraint of *transformers* to avoid conflict.

5. **Context length issues**
   - **Repository**: *gpt-researcher*

- **Link to the Issue**: https://github.com/assafelovic/gpt-researcher/issues/1196
- **Link to the PR**: https://github.com/assafelovic/gpt-researcher/pull/1195
- **Issue Description**: Exceed maximum context length error in generate_report: Expected a string with maximum length 1048576, but got a string with length 1304783 instead.
- **Fix Strategy**: Limit context to 25k words (with safety margin) by trimming older records while keeping recent and relevant ones.

6. **Prompt-related issues**

- **Repository**: *gpt-researcher*
- **Link to the Issue**: https://github.com/assafelovic/gpt-researcher/issues/1100
- **Link to the PR**: https://github.com/assafelovic/gpt-researcher/pull/1101
- **Issue Description**: "Introduction" and "Conclusion" sections remain in English even when LANGUAGE is set to a different language (e.g., "japanese") in the configuration.
- **Fix Strategy**: Update the prompts for "Introduction" and "Conclusion" generation to include language specification instructions.

## F.5 Workflow issues

1. **Workflow issues**

- **Repository**: *CrewAI*
- **Link to the Issue**: https://github.com/crewAIInc/crewAI/issues/1463
- **Link to the PR**: https://github.com/crewAIInc/crewAI/pull/1531
- **Issue Description**: Execution fails for steps with multiple preceding parallel steps.
- **Fix Strategy**: Modify the asynchronous listening mechanism to ensure that subsequent steps can proceed smoothly after the preceding parallel steps are completed.

## F.6 Utility issues

1. **Utility implementation issues**

- **Repository**: *SWE-agent*
- **Link to the Issue**: https://github.com/SWE-agent/SWE-agent/issues/362
- **Link to the PR**: https://github.com/SWE-agent/SWE-agent/pull/497
- **Issue Description**: In the *React* frontend framework, the default value of a textbox is supposed to update based on another selected dropdown item, but this dynamic binding is not functioning as intended.
- **Fix Strategy**: Use `useEffect` to monitor the change of the dropdown item, and when it changes, reset the textbox to display the default value.

2. **Utility dependency issues**

- **Repository**: *autogen*
- **Link to the Issue**: https://github.com/microsoft/autogen/issues/1436
- **Link to the PR**: https://github.com/microsoft/autogen/pull/1437
- **Issue Description**: Running *pytest* with the latest version 8.0.0 released an hour ago is not working.
- **Fix Strategy**: Limit version of *pytest* to under 8.0.0

3. **Utility configuration issues**

- **Repository**: *CrewAI*
- **Link to the Issue**: https://github.com/crewAIInc/crewAI/issues/1270
- **Link to the PR**: https://github.com/crewAIInc/crewAI/pull/1316
- **Issue Description**: When loading a variable from the yaml environment, the crewai library tries to load it in gbk encoding as soon as Chinese is present in it, but my yaml file is utf-8. You want to add a measure to detect file encoding when loading yaml files.
- **Fix Strategy**: Explicitly specify UTF-8 encoding when reading the YAML file.

Table 7: Overall results of SE agents on AGENTISSUE-BENCH with task difficulty

| SE Agent | LLM | Plausibly resolved% | Correctly resolved% | Localization % | |
| | | | | File-level | Function-level |
|---|---|---|---|---|---|
| Agentless | GPT-4o | 11.46 | 3.51 | 27.25 | 12.08 |
| | Claude-3.5-S | 11.32 | 3.77 | 26.82 | 16.88 |
| AutoCodeRover | GPT-4o | 6.84 | 1.40 | 21.62 | 14.37 |
| | Claude-3.5-S | 11.58 | 4.33 | 25.21 | 18.46 |
| SWE-agent | GPT-4o | 0.70 | 0.70 | 11.26 | 3.85 |
| | Claude-3.5-S | 2.11 | 2.11 | 9.15 | 6.40 |

Table 8: Pass@1 vs. Pass@3 on AGENTISSUE-BENCH

| Pass@k | GPT-4o (%) | | | Claude-3.5-S (%) | | |
| | Agentless | AutoCodeRover | SWE-agent | Agentless | AutoCodeRover | SWE-agent |
|---|---|---|---|---|---|---|
| Pass@1 | 6.00 | 2.00 | 2.00 | 4.00 | 6.00 | 2.00 |
| Pass@3 | 6.00 | 2.00 | 2.00 | 4.00 | 6.00 | 2.00 |

# G   Comparison among SE agents with various metrics

We further evaluate the fixing capabilities of SE agents by considering task difficulties. In particular, we consider the utility errors as of low severity/difficulty while the other agent-specific issues as of high severity/difficulty. Table 7 presents the weighted resolution rate (0.2 for utility errors and 0.8 for other agent-specific errors) among SE agents. Overall, we could observe similar findings between weighting and non-weighting results, including (1) limited capabilities of agents in agent issue resolution, (2) the superiority of AutoCodeRover with Claude-3.5-S, and (3) outperformance of Claude-3.5-S over GPT-4o.

Table 8 presents the pass@1 and pass@3 over one run. Overall, we could observe consistent trends on these metrics as our current findings, including (1) overall limited capabilities of agents in agent issue resolution, (2) the superiority of AutoCodeRover with Claude-3.5-S, and (3) outperformance of Claude-3.5-S over GPT-4o.

# H   Extended AGENTISSUE-BENCH

Table 9 presents the results of SE agents on the 20 more issues [6] that are additionally reproduced after the paper submission time. Overall, we could observe similar trends on these additional issues that SE agents exhibit limited capabilities of resolving agent issues (i.e., up to 5% correct resolution rate).

Table 9: Results of SE agents on additional issues

| SE Agent | LLM | Plausibly resolved% | Correctly resolved% | Localization % | |
| | | | | File-level | Function-level |
|---|---|---|---|---|---|
| Agentless | GPT-4o | 5.00 | 0.00 | 10.00 | 6.67 |
| | Claude-3.5-S | 5.00 | 0.00 | 12.50 | 6.67 |
| AutoCodeRover | GPT-4o | 5.00 | 0.00 | 12.50 | 5.83 |
| | Claude-3.5-S | 5.00 | 5.00 | 20.83 | 11.25 |
| SWE-agent | GPT-4o | 0.00 | 0.00 | 0.00 | 0.00 |
| | Claude-3.5-S | 0.00 | 0.00 | 0.00 | 0.00 |

