# OpenReview forum: "Can Agent Fix Agent Issues?"
_NeurIPS.cc/2025/Conference — NeurIPS 2025 poster_

### Official Review · Reviewer_o2xH · 2025-06-20

**Clarity:** 3
**Significance:** 2
**Originality:** 1
**Rating:** 5
**Confidence:** 4

**Summary:**

This paper presents a comprehensive study on the challenges of maintaining and resolving issues in LLM-based agent systems. The authors first conduct an empirical study by manually analyzing 201 real-world GitHub issues from 16 popular agent systems, resulting in the first taxonomy of agent issues. This taxonomy categorizes problems into six main areas: Incompatibility with LLM providers, Tool-related issues, Memory-related issues, LLM operation issues, Workflow issues, and Utility issues, revealing that many issues are unique to the agent paradigm and distinct from those in traditional software.

Building on this analysis, the paper introduces AGENTISSUE-BENCH, a new benchmark for evaluating the automated resolution of agent-related issues. The benchmark consists of 50 reproducible issue resolution tasks, each packaged in a Docker environment with failure-triggering tests. The authors then evaluate three state-of-the-art Software Engineering (SE) agents (Agentless, AutoCodeRover, and SWE-agent) on this benchmark. The results show that these agents have limited effectiveness, with correct resolution rates ranging from only 3.33% to 12.67%. This performance is significantly lower than their success rates on traditional software bugs, highlighting the unique challenges posed by agent systems and underscoring the need for specialized SE agents.

**Questions:**

1. Your taxonomy of agent issues shares conceptual overlap with prior work on bugs in LLM-integrated systems, such as [43]. Could you elaborate on the key distinctions and novel insights your taxonomy provides by offering a more direct comparison with the defect classes identified in previous studies?

2. In your evaluation, you report the average resolution rate over three runs. Could you justify this metric choice over a more standard one like Pass@k, which is common for stochastic tasks, and clarify how an issue was determined to be "resolved" for the Venn diagrams in Figure 5?

3. Given that your findings reveal current SE agents are ill-equipped to handle agent-specific issues, what novel methods or architectural changes do you hypothesize are needed to build agents that can effectively tackle the complex LLM operation, memory, and workflow problems identified in your benchmark?

**Ethical Concerns:**

["NO or VERY MINOR ethics concerns only"]

**Final Justification:**

The authors addressed my concerns:
- discussed in more details the comparison with related work
- Clarified the metrics used
- Discussed the future works

**Limitations:**

This paper provides a valuable and timely investigation into the maintenance challenges of the rapidly emerging paradigm of LLM-based agent systems. The primary strength of this work is the creation and public release of AGENTISSUE-BENCH, the first reproducible benchmark specifically designed for agent issue resolution. The amount of effort invested—500 person-hours to create 50 reproducible tasks—is substantial. This benchmark, complete with executable environments and failure-triggering tests, represents a significant resource for the community. Furthermore, the empirical study and the resulting taxonomy of agent issues provide a structured understanding of the types of bugs and feature requests that are common in these systems, which is a valuable contribution in itself.

Despite its strengths, the paper has a few weaknesses that detract from its overall impact, particularly concerning its framing as a main track research paper, the novelty of its taxonomy, and certain aspects of its evaluation methodology.

## Low Significance for main research track
The paper has low research significance for a main research track, and is better suited for a dataset/benchmark track. While the authors spend a significant amount of time and effort curating this dataset, there is not much novelty in the core methodology. The authors investigate a specific set of bugs, they reproduce existing techniques on these bugs, and do not introduce any significant novelty in terms of new algorithms or approaches for fixing these agent-specific issues. The main contribution is the benchmark and the empirical findings from applying existing tools. While this is a very valuable contribution to the field, it is primarily a resource paper. A main research track typically expects novel techniques or fundamental theoretical insights, whereas this work's core is an empirical study and the creation of an evaluation suite.

## Taxonomy comparison with existing ones
The presented taxonomy shares some commonalities with existing ones that focus on the integration of LLMs within software systems. As a matter of fact, the main distinction of these bugs with other traditional software systems is the presence of LLMs. The paper mentions a related work [43] by Shao et al. which also catalogs bugs in LLM-integrated systems. However, the authors do not provide a detailed comparison. For instance, the "Incompatibility with LLM providers" and "LLM operation issues" categories in this paper appear to overlap significantly with the defect classes identified in [43]. A more thorough discussion comparing and contrasting the two taxonomies would have better highlighted the unique contributions of this work. Without this comparison, the novelty of the taxonomy is somewhat reduced, as it's unclear how much it extends beyond previously identified integration problems.

## Evaluation
The evaluation methodology could be strengthened.Using the average of three runs to report resolution rates is a reasonable approach to handle the non-determinism of LLMs, but a metric like Pass@k, which is standard in code generation and related fields, would probably represent a better metric. It is also unclear how the Venn diagrams in Figure 5 were produced given the three runs they performed. Are any issues solved in any of the three runs counted? For an issue to be counted as "solved" by an agent in the diagram, does it need to be solved in all three runs, or just one? This ambiguity makes it difficult to interpret the complementary strengths of the different SE agents. Clarifying the criterion for inclusion in the Venn diagrams is essential for the reproducibility and clear interpretation of these results.

**Quality:**

3

**Strengths And Weaknesses:**

# Strengths
- The paper introduces AGENTISSUE-BENCH, a high-quality and reproducible benchmark for agent issue resolution.
- It presents the first taxonomy of agent issues, which is based on an extensive manual analysis of real-world examples.
- The work provides significant results that reveal the limitations of current SE agents in handling agent-specific problems.

# Weaknesses
- The paper's primary contribution is a benchmark, which lacks the significant methodological novelty for a main research track.
- Its taxonomy is not sufficiently compared to existing work, which diminishes its perceived novelty.
- The evaluation relies on non-standard metrics, and the reporting of some results lacks clarity.

---

> ### Author Rebuttal · Authors · 2025-07-31
>
> **1:"Taxonomy comparison with [43]”**
>
> [43] only centers on LLM+RAG systems whereas we target a broader range of LLM-based agents.  Such distinctions in scope and focus that lead to notable differences in our taxonomies:
>
> * **1. Broader Coverage of Tool-Related Issues:**   [43] only includes issues on retrieval tools, due to its focus on LLM+RAG systems. However, agents often incorporate a wider range of tools beyond RAG, such as file I/O tools, command-line interfaces, and web APIs. Our taxonomy captures a more comprehensive set of tool-related issues, including tool dependency, configuration, and implementation, which are NOT covered in [43] but frequently occur in real-world agents.
>
> * **2. Finer-Grained Categorization of Memory Issues:** [43] includes only a single category for inefficient memory problems, whereas our taxonomy provides a more detailed breakdown of memory-related issues. We additionally cover issues on memory initialization, content errors, and memory dependencies, which are critical in agents but missing in [43].
>
> * **3. Inclusion of API and Model Binding Issues:** Our taxonomy explicitly includes issues caused by dependencies on external APIs or model bindings, e.g., agents often support switching between LLMs, which can introduce binding or invocation failures. The entire issue category is common in practice but missing in [43].
>
> * **4. Broader Workflow Issue Categories:** [43] only identifies a single workflow-related issue: "Out-of-sync LLM downstream tasks". However, workflow failures in agent systems can also include unintended suspensions, incorrect execution sequences, and broken workflows. Our taxonomy reflects this broader set of workflow concerns.
>
> * **5. Component-Oriented vs. Cause-Oriented Organization:** Our taxonomy is organized by system components (e.g., LLM, tool, memory), enabling clearer mapping of issues to the corresponding agent parts. In contrast, [43] uses a cause-based structure, which can lead to overlapping or ambiguous categories  (e.g., “Unclear context in prompt” vs. “Lacking restrictions in prompt”).
>
> **2: “Justify the metric choice? clarify how an issue was determined to be "resolved" for Figure 5?”**
>
> - *Why use average resolution rate*. First, we’d like to clarify that our used metric (average resolution rate over three runs) is essentially pass@1, but we present average pass@1 over three runs to mitigate the randomness from LLMs. In fact, all the existing work[1-4]  on automated issue resolution reports pass@1, as in practice developers often check only one patch. Therefore, our metric is actually in line with the common practice of existing work, but we perform more rigorous evaluation by repeating our experiment three times.
>
>   Moreover, based on reviewers’ suggestions, we further present the pass@1 and pass@3 over one run in the following table. Overall, we could observe consistent trends on these metrics as our current findings, including  (1) overall limited capabilities of agents in agent issue resolution, (2) the superiority of AutoCoveRover with Claude-3.5, and  (3) outperformance of Claude-3.5 over GPT-4o. We’ll include these discussions into our paper.
>
> |  | w\Gpt-4o    |  |  | w\Claude-3-5-sonnet |   |  |
> |---|---|---|---|---|---|---|
> |  | **Agentless** | **SWE-Agent** | **AutoCodeRover** | **Agentless** | **SWE-Agent** | **AutoCodeRover** |
> | **Pass@1**  | 6.00%  | 2.00% | 2.00%   | 10.00%  | 8.00%| 14.00%  |
> | **Pass@3**  | 10.00%  | 6.00% | 8.00% | 10.00%  | 8.00% | 14.00% |
>
>
> - *How an issue was determined to be "resolved" for Venn diagram Figure 5?*
>  For an issue to be included in the Venn diagram in Figure 5, it needed to be solved by the agent in at least one of the three experimental runs. The reason behind this “best-of-three” approach was to capture the agent’s capability and effectively visualize the complementary strengths and overlaps of each SE agent. We’ll clarify this in our paper.
>
> **3. “What novel methods or architectural changes are needed?”**
>
> Based on our study, there can be several architectural improvements towards building more effective SE agents for diagnosing and resolving issues in LLM-based agent systems:
>
> * **Adding a knowledge base on agent-needed external resources.** As our study shows that SE agents fail to fix external resource-related (e.g., tools, dependencies, or LLM providers) issues, it indicates that SE agents need to be enhanced with more up-to-date and evolving knowledge on how to use these agent-needed external resources. Thus, one architectural improvement is to add and maintain an evolving knowledge base on all agent-related external resources (e.g., built on the API documentation, release notes, or historical issues), and then integrating with retrieval-augmented generation could help SE agents better reason and diagnose the agent issues related to external resources.
>
> * **Training better SE agents with instances and trajectories collected from AgentIssue-bench.**  Recent work[10] has shown that task instances and agent trajectories can train more powerful SE agents for *traditional* software issue resolution (e.g., SWE-bench). Our benchmark further fills the gap by providing training data specifically on the emerging software system – agent systems. Specifically, as our work provides executable environments and tests of buggy/fixed agents, we can collect massive instances and trajectories (e.g., agent-environment/tool interaction trajectories) for fine-tuning more powerful SE agents specifically for agent issue resolution.
>
> * **Adding a dynamic analysis component in SE agents.** As our study shows the limited *localization accuracy* of SE agents in agent issue resolution, it indicates that one big challenge for agent issue resolution lies in the huge gap between the issue description and root causes. However, existing SE agents only analyze *static information* (i.e., error symptoms in  issue descriptions) for issue resolution, leaving massive important runtime information untapped. To tackle this limitation, SE agents should add a dynamic analysis component, to utilize the runtime and intermediate outputs during agent operation (e.g., trajectories, fine-grained outputs of each key step like tool invocation, memory updating). Such a dynamic analysis can provide more informative hints for accurate issue localization and patch generation.
>
> We believe these topics above would  provide actionable guidelines for future work towards building more powerful SE agents. We’ll include these discussions  into our paper.
>
> **4. “ While this is a very valuable contribution to the field, a main research track typically expects novel techniques, whereas this work's core is an empirical study...”**
>
> We appreciate your recognition of the value of the benchmark and empirical findings.  We believe the manuscript fits better in the main NeurIPS conference because it goes beyond mere benchmarks but also presents both the taxonomy of agent issues and important findings：
>
> * **New problem for automated agent issue resolution**: This work makes the first attempt to formalize and define the automated agent issue resolution problem. Given the rapid evolution and increasing adoption of agent systems, we believe it is critical to bring attention to the issue resolution problem of this emerging software category.
>
> * **Empirically grounded taxonomy of agent issues**:  We perform a rigorous open-coding analysis to develop a comprehensive taxonomy of common agent issues. As responded in the first comment, we believe our taxonomy provides novel and different scope from existing work. We believe our taxonomy enhances the community’s conceptual understanding of the problem space and provides a structured foundation for future research on agent system maintenance.
>
> * **Novel findings revealing limited capabilities of SOTA agents in solving agent issues:** We present an extensive empirical study to reveal the limited capabilities of SOTA  SE agents in agent issue resolution, with diverse experimental settings and analysis. As responded in the third comment, these findings further indicate practical guidelines to build more powerful SE agents.
>
> In summary, this paper highlights a novel challenge: issue resolution in agent-based systems by SE agents. It provides a clear taxonomy of unique failure scenarios and exposes key challenges faced by SE agents.  These insights go beyond traditional benchmarking and offer broader implications for the design and evaluation of agent systems, which will be of broad interest to the NeurIPS community. In fact, the past editions of main Neurips conferences have also been accepting empirical studies on exploring the capabilities of LLMs/agents[5-9].
>
> ----
> Reference
>
> [1] Tao, Wei, et al. "Magis: Llm-based multi-agent framework for github issue resolution." Neurips 2024.
>
> [2] Zhang, et al. "Autocoderover: Autonomous program improvement." ISSTA 2024.
>
> [3] Yang et al. "Swe-agent: Agent-computer interfaces enable automated software engineering." Neurips 2024
>
> [4] Ma  et al. "Alibaba lingmaagent: Improving automated issue resolution via comprehensive repository exploration." arXiv 2024.
>
> [5] Deng, Yinlin, et al. "Can LLMs implicitly learn numeric parameter constraints in data science APIs?." Neurips main conference (2024)
>
> [6] Ma, Yiwei, et al. "I2ebench: A comprehensive benchmark for instruction-based image editing." Neurips main conference (2024)
>
> [7] Ho, Sy-Tuyen, et al. "Vision Transformer Neural Architecture Search for Out-of-Distribution Generalization: Benchmark and Insights." Neurips main conference (2024)
>
> [8] Duan et al. "Gtbench: Uncovering the strategic reasoning capabilities of llms via game-theoretic evaluations." Neurips main conference (2024)
>
> [9] Zeng et al. "Mr-ben: A meta-reasoning benchmark for evaluating system-2 thinking in llms." Neurips main conference (2024)
>
> [10]  Pan et al.  Training Software Engineering Agents and Verifiers with SWE-Gym. ICML 2025.

---

> > ### Author Response · Authors · 2025-08-05
> >
> > Dear Reviewer o2xH,
> >
> > Sorry for disturbing you, but we just heard from the program chairs that we shall remind reviewers to initiate the discussion. Please kindly let us know if our response has addressed your questions and if you have any remaining concerns.  Thank you in advance and we really look forward to further communicating with you!

---

> > > ### Comment · Reviewer_o2xH · 2025-08-06
> > >
> > > I appreciate the Authors' detailed answer and additional results presented. I hope to see these information incorporated in the final version of the paper. I'm revising the score as accept.

---

> > > > ### Author Response · Authors · 2025-08-09
> > > >
> > > > Dear reviewer,
> > > >
> > > > Thank you so much for your feedback. We promise to include the information above in the final version of our paper. Thanks!

---

### Official Review · Reviewer_pzHu · 2025-07-01

**Clarity:** 4
**Significance:** 2
**Originality:** 2
**Rating:** 4
**Confidence:** 4

**Summary:**

The paper highlights the emerging software paradigm of "agent systems", and perform an empirical study to taxonomize issues observed in agent software repositories, highlighting that the issues are different from traditional software issues and therefore motivating the need to study automated issue resolution for this subdomain of software engineering.

Next, the paper proposes AgentIssue-Bench, a (N=50) benchmark for software-engineering agents, consisting of tasks to solve issues occuring in agent systems. Each task instance in the benchmark consists of executable environment packaged as docker images, along with unit tests and manually implemented and verified failure-to-pass tests. It is demonstrated that current SE agents with leading LLMs achieve a resolution rate of 3.33% to 12.67% highlighting the need for improvement in this subdomain. The authors highlight that majority issues solved pertain to dependency related issues, whereas most tasks related to agent capabilities remain unsolved.

**Questions:**

Can the authors please clarify how the semantic equivalence was checked to differentiate "plausible correct" and "correct" issue resolutions?

Can the authors please clarify why is the function level localization percentage lower than plausibly resolved percentage? Does this indicate that there are issues where the LLM generated solution was able to solve the testcases even without correctly localizing the bugs? If so, does this indicate issues with the quality of unit tests (For example, SWEBench-Verified creates a rubric to identify good qualities in unit-tests for testing SE agent capabilities).

**Ethical Concerns:**

["NO or VERY MINOR ethics concerns only"]

**Final Justification:**

The authors have clarified the discrepancy in function-level localization and plausibly resolved percentage, highlighted that to tackle the time-dependency issue I raised, they have maintained docker containers checkpointing repository states. Further, they have highlighted the collection of more tasks, along with the intent to continuously update the tasks, addressing my concern about the time-dependency of the benchmark.

**Limitations:**

yes: Specifically, the authors highlight the dataset size (n=50), as well as being restricted to 16 agentic systems

**Quality:**

3

**Strengths And Weaknesses:**

I really appreciate the overall problem identified by the paper, which is about the emerging class of LLM-integrated software that handles non-determinism, yet aiming to provide a coherent experience to its users. The taxonomy study applied to this specific subdomain is useful and valuable. Further, the constructed benchmark has tests manually verified by human annotators, which is great for benchmark quality. The authors also make a distinction between "plausible correct resolution" and "correct resolution", which is a step in the right direction for SE agent benchmarking.

However, I have the following concerns with the benchmark:
As the authors highlight in the paper, reproducing issues in agent repositories is difficult, because, "External resources (e.g., agent-invoked tools, dependent libraries, or LLM 228 providers) may have changed since the issue was reported ...".  Precisely because of this reason, the core contribution of the paper, which is the benchmark and its unit-tests, may not hold in a few months time. As an example, the authors highlight in Figure 6, that "o1-preview", and "o1-mini" do not support the stop keyword. While the fix is pragmatic (to include a parameter), the issue itself may become irrelevant: "o1" model, from the same series does support the parameter. This implies that the issues included in the benchmark are very time dependent on the model providers behaviours (which is not fixed, and can change at any time without notice), and therefore, not capture the performance/reasoning ability of the models, but rather their time-specific awareness of the current state of LLM APIs.
2. While the paper is well motivated, and the benchmark curation approach is sound, the authors claim through the taxonomy that the type of issues occuring in agentic repositories are such that they typically do not apply to traditional software. However, going through Appendix F, where the authors list an example issue from every category, I present the following annotation (from my understanding) of the underlying issue captured in the highlighted example:

```
F.1.1: Dependency
F.1.2: Dependency
F.1.3: Incompatible parameters to external LLM providers
F.2.1: Dependency
F.2.2: Environment variable related configuration
F.2.3: File read/encoding
F.2.4: Information about how to call wikipedia API
F.3.1: Software configuration issue
F.3.2: SambaNova API specific
F.3.3: Dependency changed the public API and those changes not migrated
F.4.1: Environment variable related configuration
F.4.2: TypeError
F.4.3: Gemini's content policy rejects specific prompts. Add new exception to fix it.
F.4.4: Dependency
F.4.5: Context Window Management
F.4.6: Changing model behaviour using prompt
F.5.1: Control flow issue
F.6.1: Frontend/UI
F.6.2: Dependency
F.6.3: File encoding
```

Out of the 20 highlighted, 6/20 pertain to dependency upgrades, 4/20 are related to reading files and environment variables, 1/20 is a UI issue, and so on. These are very typical of traditional software. So I believe that while the stated goal of the paper is to build a benchmark to test SE agent's capabilities in solving agent-specific issues, most of the issues present in the benchmark do not actually capture this capability. From my reading, only 4/20 of the highlighted issues actually pertain to LLM-specific problems.
3. Benchmark size: The benchmark size is 50, out of which 20 are described in the appendix itself and have issues as highlighted above.

---

> ### Author Rebuttal · Authors · 2025-07-31
>
> We thank the reviewer for the constructive feedback.
>
> **1.“clarify how the semantic equivalence was checked to differentiate ‘plausible correct’ and ‘correct’  issue resolutions?”**
>
> We conduct a structured manual evaluation process to check semantic equivalence. Two participants independently review each plausible patch by comparing it to the golden patch (i.e., developer-committed), focusing on whether the semantics of the patch fully resolve the underlying issue as intended and do not introduce other functional or semantic errors. If both reviewers agree that the patch was semantically equivalent and correctly resolved the issue, it is labeled as correct. If there was a disagreement between the two reviewers, we brought in a third participant as an adjudicator. The final label was determined only after all three reviewers reached a consensus. This multi-step adjudication process ensured consistency and rigor in our evaluation of patch correctness.
> One example is autogen-1174, AutoCodeRover’s generated patch successfully passed the failure-triggering test, qualifying it as plausibly resolving the issue. However, during the manual review, we found that the generated patch incorrectly discarded a “tool_responses”: tool_returns key-value pair that the ground truth patch explicitly preserves. The developer’s ground truth patch is designed to return “role” key, “tool_responses” key, and “content” key. The generated patch alters this by removing the “tool_responses” key, returning only the “role” and the “content”. Following our methodology, this failure to replicate the full informational output of the ground truth patch means the two solutions are not semantically equivalent. Therefore, the generated patch was classified as a plausible correct but not correct. We will include the description of our check process in our paper.
>
> **2: “Why the function-level localization percentage is lower than the plausibly resolved percentage? ”**
>
> Great point. Function-level localization percentage is lower, as we exclude the issues that are not localized within a function. For example, some dependency-related issues are fixed by modifying configuration files (e.g., requirements/runtime.txt), where there is no function at all. These cases are not included in calculating function-level localization accuracy. If we also exclude these cases for calculating plausible fix rate, the plausible fix rate would be lower than function-level localization rate.
> Such a problem does not exist for file-level localization. Therefore, all the file-level localization percentage is higher than plausible fix rate, indicating the quality of our tests. We’ll clarify this in our paper.
>
> **3.“...issues included in the benchmark are very time dependent on the model providers behaviour (which is not fixed, and can change at any time without notice), and therefore, not capture the performance/reasoning ability of the models, but rather their time-specific awareness of the current state of LLM APIs. ”**
>
> * First, we’ve specifically snapshotted all the external dependencies into dockers, therefore, the majority of the external resources-related issues can stay with long-term reproducibility. Moreover, we plan to continuously maintain and monitor our benchmark so that the evolution of provider APIs would not invalidate our benchmark. For example, for the issues that are unrelated to provider APIs, they can always stay reproducible by using the latest APIs (e.g., termly running tests can automatically detect unreproducible issues and then alert us to fix it accordingly). The only exception is for provider API-related issues, although we cannot control their reproducibility, such issues only account for a small portion  (e.g., 2.49% for incompatible parameters to LLM providers) in our benchmark, thus not threatening the overall usability of our benchmark. If the reviewer has any good suggestions to maintain the long-term reproducibility of provider API-related issues, please let us know and we’re happy to incorporate it to our benchmark.
>
> * As our benchmark is mined from real-world agents, we consider it reflects the actual distribution and realistic problems developers can encounter in practice. Therefore, although provider API-related issues focus on capturing the time-specific awareness of the current state of APIs, we still believe it is an important and real-world problem that should not be ignored for building agent maintenance techniques. In fact, for traditional software, recent research efforts have been dedicated to analyzing LLMs’ awareness in using deprecated external APIs[1]. Therefore, we consider to keep this issue category in our benchmark to reflect the real-world distribution of agent issues.
>
> * In addition to such provider API-related issues, our benchmark also includes many other issues that directly capture the performance/reasoning capabilities of models, such as incorrect model output handlers (capture cases where LLMs produce unexpected or ill-formed outputs), prompt-related issues (reveal instances where the model fails to correctly interpret task-specific inputs), and memory content errors (illustrate scenarios where the model's internal context handling leads to degraded performance).
>
> In summary, we consider our benchmark as of overall long-term reproducibility.
>
> **4:“...However, going through Appendix F, I present the following annotation of the underlying issue... These are very typical of traditional software. So I believe that while the stated goal of the paper is to build a benchmark to test SE agent's capabilities in solving agent-specific issues, most of the issues present in the benchmark do not actually capture this capability."**
>
> Thanks a lot for such detailed feedback on our taxonomy and we really appreciate it! However, based on the annotation, there might be some misunderstanding for which we’d like further to clarify.
>
> Our taxonomy is derived by considering both the issue symptoms and the fix strategies. While some issues may appear similar to those found in traditional software (such as dependency updates), they pose unique challenges within agentic systems. Take F.1.1 and F.1.2 as examples. Although the fix strategy is broadly described as "updating the version of the OpenAI/Anthropic library," this task is far from trivial for an SE agent. The issue descriptions often only describe the symptom, and the agent must first infer that it stems from a library version mismatch. For instance, GPT-4 Turbo is only supported in newer versions of the OpenAI library. Beyond updating the dependency, the agent must also integrate the new model into the agent system's existing framework, including designing interfaces, updating dispatch logic, and ensuring backward compatibility with existing models. This demands a deep understanding of the agent system's architecture, its abstraction layers, and the invocation patterns of various LLMs. In this context, these issues are deeply agent-specific, not simply traditional dependency problems.
>
> Similarly, F.4.2 is not a trivial TypeError. While the symptom relates to OpenAI’s NOT_GIVEN sentinel value causing a comparison failure, the patch involves a broader architectural change, moving the token_limit property from the base_model to a unified_model_type abstraction class. This reflects a deeper system-level understanding of how token limits vary across LLMs and how such properties should be organized within the agent system to support extensibility and correctness. An SE agent would need to comprehend these architectural decisions to produce a correct and sustainable fix.
>
> Additionally, beyond the appendix examples, our paper provides a quantitative breakdown of issue types in Figure 2. Notably, 72% of the benchmarked issues involve core agent components, such as LLM orchestration, memory management, and tool integration. These challenges are specific to agentic systems, arising from their tight coupling with evolving LLM APIs, dynamic tool usage, and memory mechanisms. As such, resolving these issues requires an SE agent not only to fix code but also to reason about LLM behavior, understand architectural abstractions, and ensure system-wide consistency across heterogeneous components.
>
> In summary, while some issues may resemble those found in traditional software at a surface level, the underlying resolution process reflects challenges that are uniquely agent-specific. This directly aligns with the stated goal to evaluate the SE agent’s ability to operate in the context of agentic systems.
>
> **5: “The benchmark size is 50, out of which 20 are described in the appendix itself and have issues as highlighted above.”**
>
> While given the difficulty and huge manual effort  in reproducing agent issues, we’re only able to reproduce 50 issues as the first version of our benchmark by the submission time. However, we plan to and are also currently working to extend our benchmark in the long term. In fact, we have dedicated effort since our initial submission to expand AgentIssue-Bench by additionally including 31 more issues, most of which are agent-specific and not utility-related ones; and on the additional issues we observe similar trends that SE agents have limited capabilities of resolving agent issues (e.g., the best agent AutoCodeRover only fixes 2 issues).  We'll discuss the results in our paper and also build a leaderboard website for continuous update.
>
> ----------
> Reference:
>
> [1] LLMs Meet Library Evolution: Evaluating Deprecated API Usage in LLM-based Code Completion. Wang et al. ICSE 2025

---

> > ### Author Response · Authors · 2025-08-04
> >
> > Dear Reviewer pzHu,
> >
> > Thanks again for your reviewing effort and feedback for our work. Please kindly let us know if our response has addressed your questions and also we would be more than happy  to provide any more clarification for the additional questions you might have. Thank you!

---

> ### Comment · Reviewer_pzHu · 2025-08-06
> **Acknowlegement and score increase**
>
> Dear Authors,
>
> I apologize for the confusion caused on my end. I had read your responses previously, and correspondingly, increased my rating with the following edit in the original review: "The authors have clarified the discrepancy in function-level localization and plausibly resolved percentage, highlighted that to tackle the time-dependency issue I raised, they have maintained docker containers checkpointing repository states. Further, they have highlighted the collection of more tasks, along with the intent to continuously update the tasks, addressing my concern about the time-dependency of the benchmark."

---

> > ### Author Response · Authors · 2025-08-09
> >
> > Dear reviewer,
> >
> > Thank you so much for your feedback. We will definitely to include the information above in the final version of our paper. Thanks!

---

### Official Review · Reviewer_WCGa · 2025-07-03

**Clarity:** 3
**Significance:** 3
**Originality:** 3
**Rating:** 4
**Confidence:** 3

**Summary:**

The paper derives a six‐category taxonomy from 201 real GitHub issues in LLM-based agents, then builds AgentIssue-Bench, a 50-task Dockerized benchmark of reproducible bugs and patches.
The authors evaluated three SE agents (Agentless, AutoCodeRover, SWE-agent) with GPT-4 and Claude, finding only 3.33–12.67% successful fixes.

**Questions:**

How do the authors plan to grow AgentIssue-Bench beyond these 50 scenarios? Could parts of the Docker/test setup be automated, or could the community help contribute new tasks?

What concrete steps do you take to keep each task reproducible as LLM outputs and external tools change? Do you pin random seeds, snapshot dependencies, or filter outputs?

Have the authors thought about varying task difficulty or severity? For instance, would an “LLM operation” bug count differently than a simple utility error, and would weighting by complexity alter your evaluation?

It would be beneficial if the authors can provide thoughts on why this manuscript fit better to the main NeurIPS conference, instead of the benchmark and dataset track of NeurIPS.

**Ethical Concerns:**

["NO or VERY MINOR ethics concerns only"]

**Final Justification:**

My questions are majorly addressed by the authors' responses.

**Limitations:**

Yes there is an explicit limitation section in the manuscript.

**Quality:**

3

**Strengths And Weaknesses:**

Strengths

The manuscript is well-written and easy to follow.

The authors mine over 200 real-world GitHub issues to produce the first comprehensive taxonomy of error modes unique to LLM-based agent systems.

The AgentIssue-Bench provides a robust, one-click platform for evaluating agent debugging performance.

The evaluation has both quantitative success rates and qualitative breakdowns by issue category.

---

Weaknesses

With only 50 tasks selected from 201 identified issues, the benchmark’s limited scale may not fully represent the diversity of agent failures or support strong statistical claims.

Building each scenario demanded 500 person-hours of manual effort. The manuscript lacks a clear pathway for automating or streamlining this process as the benchmark grows.

Although the authors acknowledge challenges around LLM nondeterminism and evolving external dependencies, they didn't offer concrete strategies to ensure long-term reproducibility.

---

> ### Author Rebuttal · Authors · 2025-07-31
>
> We thank the reviewer for the recognition and constructive feedback for our work.
>
> **1.“How do the authors plan to grow AgentIssue-Bench beyond these 50 scenarios? Could parts of the Docker/test setup be automated, or could the community help contribute new tasks?”**
>
> - *What steps can be automated.* The following table summarizes the key steps for agent issue reproduction and to what extent they can be automated. As mentioned by the reviewer, indeed some key steps can be fully/partially automated, except the step *“failure-triggering test generation”*  which stands as the major bottleneck for automating the agent issue reproduction pipeline.  In fact, failure reproduction has been an open challenge in software engineering, and even for traditional software, recent agents could only reproduce <13% failure tests[1].
>
>   |Step| Content| Automation| Challenges|
>   |----|-----|-----|--------|
>   (1) Agent set-up| Configure and run agents. | Medium | Although it's feasible to automatically configure and run agents via scripts, there are often some challenging cases on setting missing/incorrect dependencies or user-privacy information (e.g., API keys) that require additional manual fixing effort.|
>   (2) Failure-triggering test generation| Generate test code to trigger the same failure as the issue description. | Low | Automated failure test reproduction has been an open challenge and is far beyond practical usage.|
>   (3) Reproduction verification| Check if the failure-triggering tests fail on the buggy version but pass on the correct version.| Full | This step is fully automated via scripts.|
>   (4) Dockerization| Create and push final Docker images to repository. | Full |  This step is fully automated via scripts.|
>
>
> - *Continuous and long-term extension.* To continually grow AgentIssue-Bench, a practical solution is to *standardize a semi-automated pipeline* to balance the manual effort and benchmark quality, including (1) fully automated steps (e.g.,  reproduction verification and dockerization), (2) human-dominated steps (e.g., failure-triggering test generation), and (3) human-enhanced steps (e.g., agent set-up). Actually, we’ve already been working on implementing such a pipeline, so as to support the long-term extension of our benchmark. Based on the semi-automated pipeline, *we have dedicated continuous effort to expand AgentIssue-bench by additionally reproducing 31 more issues* since the paper submission DDL; and on the additional issues we observe similar trends that SE agents have limited capabilities of resolving agent issues (e.g., the best SE agent AutoCodeRover only fixes 2 issues).
>
> * *How the community helps to contribute.* As an open-source benchmark, we welcome all the pull requests from users if they report new agent issues that satisfy our quality requirements. In addition, we’re open to integrating the automatic techniques/tools from the community for certain steps (e.g., failure-triggering test generation) to optimize our benchmark construction pipeline.
>
> **2.“What concrete steps do you take to keep each task reproducible as LLM outputs and external tools change? Do you pin random seeds, snapshot dependencies, or filter outputs?”**
>
> We’ve adopted the following steps to ensure reproducibility:
> * *Fixed randomness parameter and multi-round verification mechanism.* To mitigate nondeterminism introduced by LLMs, we fix parameters that control the randomness of LLM output (e.g., temperature = 0)  to reduce variability in outputs; additionally,  we run each task multiple times to filter out the cases with inconsistent LLM behaviors (as mentioned in line 218).
> * *Snapshot external dependencies in Docker.* We archive the external dependencies of each issue into an executable Docker,  which serves a fixed execution environment (i.e., including repository state and third-party dependencies), enabling long-term reproducibility.
>
> * *Long-term maintenance for our benchmark.* While with the above steps, there might still be a small number of cases becoming non-reproducible given the changes in LLM provider APIs (which cannot be snapshot and controlled). Therefore, we plan to long-term maintain our benchmark by regularly automatically running each issue to check their reproducibility – we will remove those no longer reproducible ones and continuously include new reproducible issues into the benchmark via our semi-automated pipeline.
>
> We’ll include the discussion above in our paper.
>
>  **3.“Have the authors thought about varying task difficulty or severity? For instance, would an “LLM operation” bug count differently than a simple utility error, and would weighting by complexity alter your evaluation? ”**
>
> Great point! Following the reviewer’s suggestion, we consider utility errors as of low severity/difficulty while the other agent-specific issues can be considered as of high severity/difficulty. The following table presents the weighted resolution rate (0.2 for easy tasks and 0.8 for hard tasks).  Overall, we could observe similar findings between weighting and non-weighting results, including (1) overall limited capabilities of agents in agent issue resolution, (2) the superiority of AutoCodeRover with Claude-3.5, and  (3) outperformance of Claude-3.5 over GPT-4o. We’ll include these discussions into our paper.
>
>
> |                      | w/Gpt-4o                        |                             |                           | w/Claude-3-5-sonnet              |                             |                             |
> |----------------------|----------------------------------|-----------------------------|---------------------------|----------------------------------|-----------------------------|-----------------------------|
> |                      | **Agentless** | **SWE-Agent** | **AutoCodeRover** | **Agentless** | **SWE-Agent** | **AutoCodeRover** |
> | **Correct Resolved**  | 5.34%     | 2.64%     | 3.86%         | 8.04%     | 6.35%     | 11.43%         |
> | **Plausible Resolved**| 17.83%    | 4.86%     | 11.96%        | 11.69%    | 6.35%     | 15.56%         |
>
> **4.”It would be beneficial if the authors can provide thoughts on why this manuscript fits better to the main NeurIPS conference, instead of the benchmark and dataset track of NeurIPS.”**
>
> We believe the manuscript fits better in the main NeurIPS conference because it goes beyond mere benchmarks but also presents both the taxonomy of agent issues and important findings on LLM/agent capabilities:
>
> * **New problem for automated agent issue resolution**: This work makes the first attempt to define and formalize the automated agent issue resolution problem. Given the rapid evolution and increasing adoption of agent systems, we believe it is critical to bring attention to the issue resolution problem of this emerging software category.
> * **Empirically grounded taxonomy of agent issues**:  We perform a rigorous open-coding analysis to develop a comprehensive taxonomy of common agent issues. This taxonomy enhances the community’s conceptual understanding of the problem space and provides a structured foundation for future research on agent system maintenance.
> * **Novel findings revealing limited capabilities of SOTA agents in solving agent issues:** We present an extensive empirical study to reveal the limited capabilities of state-of-the-art SE agents in agent issue resolution, with diverse experimental settings and analysis.
> We believe that such empirical studies on exploring the capabilities of LLMs/agents should be of broad interest to the NeurIPS community, as there have been many similar empirical studies in past main NeurIPS conferences[2-6].
>
> **5.”With only 50 tasks selected from 201 identified issues, the benchmark’s limited scale may not fully represent the diversity of agent failures or support strong statistical claims.”**
>
> While given the difficulty and huge manual effort  in reproducing agent issues, we’re only able to reproduce 50 issues as the first version of our benchmark by the submission time. However, we plan to and are also currently working to extend our benchmark in the long term. In fact, we have dedicated continuous effort since our initial submission to expand AgentIssue-Bench by additionally including 31 more issues;  and on the additional issues we observe similar trends that SE agents have very limited capabilities of resolving agent issues (e.g., the best SE agent AutoCodeRover only fixes 2 issues).  We'll discuss the results in our paper and also build a leaderboard website for continuous update.
>
> ------------
> Reference:
>
> [1] “AEGIS: An Agent-based Framework for General Bug Reproduction from Issue Descriptions” Wang et al, arXiv 2024.
>
> [2] Deng, Yinlin, et al. "Can LLMs implicitly learn numeric parameter constraints in data science APIs?." Neurips main conference (2024): 54205-54238.
>
> [3] Ma, Yiwei, et al. "I2ebench: A comprehensive benchmark for instruction-based image editing." Neurips main conference (2024): 41494-41516.
>
> [4] Zeng, Zhongshen, et al. "Mr-ben: A meta-reasoning benchmark for evaluating system-2 thinking in llms." Neurips main conference (2024): 119466-119546.
>
> [5] Ho, Sy-Tuyen, et al. "Vision Transformer Neural Architecture Search for Out-of-Distribution Generalization: Benchmark and Insights." Neurips main conference (2024): 84197-84245.
>
> [6] Duan, Jinhao, et al. "Gtbench: Uncovering the strategic reasoning capabilities of llms via game-theoretic evaluations." Neurips main conference (2024): 28219-28253.

---

> > ### Comment · Reviewer_WCGa · 2025-08-06
> >
> > Thanks the authors for their detailed rebuttal. I think my questions are well addressed.
> > Consequently, I would raise the Significance and Originality scores and keep my positive rating.

---

> > > ### Author Response · Authors · 2025-08-09
> > >
> > > Dear reviewer,
> > >
> > > Thank you so much for your feedback. We'll include the discussion above in the final version of our paper. Thanks!

---

### Official Review · Reviewer_61d1 · 2025-07-06

**Clarity:** 4
**Significance:** 4
**Originality:** 4
**Rating:** 5
**Confidence:** 3

**Summary:**

In this paper, the authors  investigates whether state-of-the-art software engineering (SE) agents can effectively resolve issues in LLM-based agent systems. Specifically, the authors manually analyze 201 real-world agent issues from 16 widely-used agent systems and categorize them into a taxonomy with 6 main categories and 20 sub-categories. deriving a taxonomy of agent issues with 6 categories and 20 subcategories. The authors find that key challenges unique to agent systems include LLM operation errors, tool dependencies, memory-related issues, and workflow anomalies. The authors also construct AGENTISSUE-BENCH , a benchmark of 50 agent issue-resolution tasks with Docker environments, failure-triggering tests, and ground-truth patches. The study uncover the need for specialized SE agents tailored to the complexities of LLM-based systems, such as handling non-deterministic LLM outputs, dynamic tool interactions, and intricate memory mechanisms.

**Questions:**

See weakness 1 and 2

**Ethical Concerns:**

["NO or VERY MINOR ethics concerns only"]

**Limitations:**

yes

**Quality:**

4

**Strengths And Weaknesses:**

strengths:

1. Comprehensive empirical study: The authors conduct an empirical study of 201 real-world GitHub issues from 16 widely used agent systems (e.g., MetaGPT, AutoGen)

2. Comprehensive category: The first systematic categorization of agent issues, derived from 201 manually analyzed real-world examples, provides valuable insights into the unique challenges

3. Benchmark is available: AGENTISSUE-BENCH addresses a major barrier in agent system research by offering a standardized, executable evaluation framework with Dockerized environments and test cases.


4. Comprehensive SOTA evaluation on the benchmark: Evaluates multiple SE agents (Agentless, SWE-agent, AutoCodeRover) with advanced LLM backbones (GPT-4o, Claude-3.5-Sonnet), demonstrating consistent limitations across tools.

5. Provide insightful thoughts towards multi-agent system design. The paper highlights the urgent need for specialized SE agents capable of addressing the interplay between LLMs, external tools, and system workflows. It provides open access to data, code, and benchmarks, fostering transparency and reproducibilit


weakness:

1. While the paper emphasizes issue diagnosis and resolution, it does not explore complementary aspects of agent system maintenance, such as preventative strategies (e.g., proactive LLM monitoring) or performance optimization (e.g., reducing tool dependency overhead). A broader perspective could offer a more holistic view of challenges and solutions.


2. Limited Discussion of Solutions to Identified Challenges: Can the authors provides some discussions about how to solve the challenges that they uncover? The paper excels at diagnosing problems and more concrete solutions are also expected.

---

> ### Author Rebuttal · Authors · 2025-07-31
>
> We thank the reviewer for the recognition and constructive feedback for our work.
>
> **1: "While the paper emphasizes issue diagnosis and resolution, it does not explore complementary aspects of agent system maintenance, such as preventative strategies (e.g., proactive LLM monitoring) or performance optimization (e.g., reducing tool dependency overhead).  A broader perspective could offer a more holistic view of challenges and solutions.”**
>
> Great point! We’d like to clarify that our current work primarily focuses on reactive issues (i.e., first-expose-then-fix), as our data source is *user-reported GitHub issues*, which inherently reflect problems that have already been encountered and reported by agent users. As a result, preventative strategies (e.g., proactive LLM monitoring) and performance optimization are not typically observable in our dataset, since these aspects are difficult to capture from *a user’s perspective*. Rather, they are more often observed from the developer’s perspective. To investigate such issues, a more suitable data source would be *internal logs from agent development and operations*, which are generally available only to industry practitioners or professional agent developers — thus beyond the scope and access of our current study. That said, we agree that these issues are important and represent promising directions for future work. We will clarify the scope and coverage of our study in the paper.
>
> **2.“Provide some discussions about how to solve the challenges.”**
>
> Great point. Based on our study, we believe there can be several architectural improvements towards building more effective SE agents for diagnosing and resolving issues in LLM-based agent systems:
> * **Adding a knowledge base on agent-needed external resources.** As our study reveals that existing SE agents fall short in fixing external resource-related (e.g., tools, dependencies, or LLM providers) issues, it indicates that existing SE agents need to be enhanced with more up-to-date and evolving knowledge on how to use these agent-needed external resources. Therefore, one architectural improvement is to add and maintain an evolving knowledge base on all agent-related external resources (e.g., built on the API documentation, release notes, or historical issues), and then integrating with retrieval-augmented generation (RAG) could help SE agents better reason and diagnose the agent issues related to external resources.
>
> * **Training better SE agents with instances and trajectories collected from AgentIssue-bench.** Recent work[1] has shown that task instances and agent trajectories can help train more powerful SE agents for *traditional* software issue resolution (e.g., SWE-bench). Our benchmark and study further fill the gap by providing training data specifically on the emerging software system – agent systems. Specifically, as our work provides executable environments and tests of buggy/fixed agent systems, we can collect massive instances and trajectories (e.g., agent-environment/tool interactive trajectories) for fine-tuning more powerful SE agents that specifically targets at agent issue resolution.
>
> * **Adding a dynamic analysis component in SE agents.** As our study reveals the limited *localization accuracy* of existing SE agents in fixing agent issues, it indicates that one big  challenge for agent issue resolution lies in the huge gap between the issue description and root causes. However, existing SE agents only analyze the *static information* (i.e., error symptoms in  issue descriptions) for issue resolution, leaving massive important runtime information untapped. To tackle this limitation, SE agents should add a dynamic analysis component, which can utilize the runtime and intermediate outputs during agent operation (e.g., trajectories, fine-grained outputs of each key step like tool invocation, memory updating). Such a dynamic analysis can provide more informative hints for accurate issue localization and patch generation.
>
> We believe these topics above would  provide actionable guidelines for future work towards building more powerful SE agents. We’ll include these discussions  into our paper.
>
> ---------------
> Reference:
>
> [1] Training Software Engineering Agents and Verifiers with SWE-Gym. Pan et al.  ICML 2025.

---

> > ### Comment · Reviewer_61d1 · 2025-08-04
> >
> > Thank you for your feedback. The authors have addressed my concerns, and I am satisfied with their response. I will keep my score as "Accept."

---

> > > ### Author Response · Authors · 2025-08-09
> > >
> > > Dear reviewer,
> > >
> > > Thank you so much for your feedback. We promise to include the discussion above in the final version of our paper. Thanks!

---

### Decision · Program_Chairs · 2025-09-17

**Decision:**

Accept (poster)

**Comment:**

This research provides a valuable empirical foundation by analyzing 201 real-world GitHub issues from 16 systems to create the first taxonomy of agent-specific issues—comprising 6 categories and 20 subcategories—and introduces AGENTISSUE-BENCH, a Docker-packaged benchmark with 50 reproducible tasks for evaluating issue resolution. Tests of three leading software engineering agents (Agentless, AutoCodeRover, SWE-agent) using GPT-4 and Claude revealed critically low success rates (3.33–12.67%), exposing their inadequacy in handling challenges such as non-deterministic outputs and dynamic tool interactions. While the paper is well-written and methodologically rigorous with a technically sound benchmark, it could be strengthened by deeper exploration of preventative measures, performance optimizations, and more substantive discussion of solutions, in addition to improving metric clarity and standardization. Despite these limitations, reviewers ultimately lean toward acceptance due to the significant empirical contribution and practical utility of the benchmark.